# HP1α is a chromatin crosslinker that controls nuclear and mitotic chromosome mechanics

Amy R Strom[1†], Ronald J Biggs[2†], Edward J Banigan[3†], Xiaotao Wang[4], Katherine Chiu[5], Cameron Herman[2], Jimena Collado[2], Feng Yue[4,6], Joan C Ritland Politz[7], Leah J Tait[7], David Scalzo[7], Agnes Telling[7], Mark Groudine[7], Clifford P Brangwynne[1], John F Marko[2,8], Andrew D Stephens[5*]

[1]Howard Hughes Medical Institute, Department of Chemical and Biological Engineering, Princeton University, Princeton, United States; [2]Department of Molecular Biosciences, Northwestern University, Evanston, United States; [3]Institute for Medical Engineering and Science and Department of Physics, Massachusetts Institute of Technology, Cambridge, United States; [4]Department of Biochemistry and Molecular Genetics, Feinberg School of Medicine, Northwestern University, Chicago, United States; [5]Biology Department, University of Massachusetts Amherst, Amherst, United States; [6]Robert H. Lurie Comprehensive Cancer Center, Feinberg School of Medicine, Northwestern University, Chicago, United States; [7]The Fred Hutchinson Cancer Research Center, Seattle, United States; [8]Department of Physics and Astronomy, Northwestern University, Evanston, United States

*For correspondence:
Andrew.stephens@umass.edu

†These authors contributed equally to this work

Competing interests: The authors declare that no competing interests exist.

**Abstract** Chromatin, which consists of DNA and associated proteins, contains genetic information and is a mechanical component of the nucleus. Heterochromatic histone methylation controls nucleus and chromosome stiffness, but the contribution of heterochromatin protein HP1α (CBX5) is unknown. We used a novel HP1α auxin-inducible degron human cell line to rapidly degrade HP1α. Degradation did not alter transcription, local chromatin compaction, or histone methylation, but did decrease chromatin stiffness. Single-nucleus micromanipulation reveals that HP1α is essential to chromatin-based mechanics and maintains nuclear morphology, separate from histone methylation. Further experiments with dimerization-deficient HP1α[I165E] indicate that chromatin crosslinking via HP1α dimerization is critical, while polymer simulations demonstrate the importance of chromatin-chromatin crosslinkers in mechanics. In mitotic chromosomes, HP1α similarly bolsters stiffness while aiding in mitotic alignment and faithful segregation. HP1α is therefore a critical chromatin-crosslinking protein that provides mechanical strength to chromosomes and the nucleus throughout the cell cycle and supports cellular functions.

## Introduction

Chromatin, which fills the nucleus, is a repository of information, but it is also a physical element that provides structure, mechanical rigidity, shape, and function to the nucleus. Heterochromatin is the stiff, compact, and gene-poor form of chromatin. Heterochromatin loss results in abnormal nuclear morphology, which is a hallmark of human disease (*Stephens et al., 2019a*; *Uhler and Shivashankar, 2018*). Increasing the amount of heterochromatin by elevating histone methylation levels can increase nuclear stiffness and restore nuclear shape and function in perturbed model cell lines and patient cells of human diseases (*Liu et al., 2018*; *Stephens et al., 2019b*; *Stephens et al., 2018*; *Stephens et al., 2017*). Chromatin stiffness also plays a key role during cell division, as mitotic

chromosome mechanics are key to the proper segregation of the genome during mitosis (*Batty and Gerlich, 2019*; *Ribeiro et al., 2009*; *Stephens et al., 2011*; *Sun et al., 2018*). Recently, it has been reported that methylated histones/heterochromatin are also a mechanical component of mitotic chromosomes (*Biggs et al., 2019*). However, in addition to methylated histones, protein 'readers' of epigenetic marks play a key role in defining heterochromatin (and euchromatin). A key histone methylation reader, Heterochromatin Protein 1α (HP1α), remains poorly characterized as to its role in controlling the mechanical properties of heterochromatin. To what degree HP1α contributes to the mechanical resistive capabilities of chromatin, how this contribution is intertwined with histone methylation, and how these result in proper nuclear and mitotic mechanics and function, are all open questions.

HP1α is a major component of constitutive heterochromatin (*James and Elgin, 1986*; *Singh et al., 1991*; *Wreggett et al., 1994*). Functionally, HP1α is a homodimer that binds to both DNA and to H3K9me$^{2,3}$ constitutive heterochromatin marks. The direct association of HP1α with H3K9me$^{2,3}$ heterochromatin and its direct binding to Suv39h1/2, the histone methyltransferase that deposits H3K9me$^{2,3}$, has led to reports that HP1α is necessary for either maintenance or establishment of histone methylation (*Bannister et al., 2001*; *Krouwels et al., 2005*).

Loss of HP1α could therefore indirectly alter chromatin mechanics by modulating histone methylation levels. Alternatively, HP1α homodimerization and/or higher-order oligomerization could directly impact mechanics through physical bridging of two chromatin fibers, resulting in crosslinking of DNA or H3K9me$^{2,3}$-marked nucleosomes (*Canzio et al., 2011*; *Cheutin et al., 2003*; *Machida et al., 2018*). Consistent with this possibility, chromatin crosslinks have been suggested to be a key element of chromatin organization and mechanics (*Banigan et al., 2017*; *Belaghzal et al., 2021*; *Lionetti et al., 2020*; *Stephens et al., 2017*). The capacity of HP1α to drive liquid-liquid phase separation (*Larson et al., 2017*; *Strom et al., 2017*) could also contribute to altered chromatin organization and mechanics, given the emerging evidence for links between phase separation and nuclear mechanics (*Shin et al., 2018*). These mechanisms could also affect mechanics in mitotic chromosomes, where HP1α is also present (*Akram et al., 2018*; *Serrano et al., 2009*). Therefore, it is now critical to determine the role of HP1α in controlling chromatin mechanics during both interphase and mitosis, as well as the functions of HP1α-mediated chromatin mechanics.

Nuclear and mitotic chromosome micromanipulation force measurements have been critical to understanding the mechanical properties of chromatin, making these techniques ideal for probing the relative roles of histone modifications and chromatin-binding proteins. Nucleus micromanipulation force measurements provide a novel capability, allowing the separation of chromatin, which dominates the initial force-response regime, from the other major mechanical component, lamin A, which dictates strain stiffening in the long-extension regime (*Stephens et al., 2017*). This two-regime force response was recently verified by the AFM-SPIM force measurement technique (*Hobson et al., 2020a*). Chromatin-based nuclear mechanics are dictated by euchromatin and heterochromatin levels, particularly through post-translational modifications of histones by acetylation or methylation, respectively (*Heo et al., 2016*; *Hobson and Stephens, 2020b*; *Krause et al., 2019*; *Liu et al., 2018*; *Nava et al., 2020*; *Stephens et al., 2019b*; *Stephens et al., 2018*; *Stephens et al., 2017*). These changes in chromatin-based nuclear mechanics can, independently of lamins, cause abnormal nuclear morphology, which is a hallmark of human disease (*Stephens et al., 2019a*). A recent high-throughput screen revealed that many key chromatin proteins also contribute to nuclear shape (*Tamashunas et al., 2020*), raising the question of the relative roles of histone modifications versus chromatin proteins such as HP1α.

Recent experimental and modeling studies suggest chromatin proteins, like HP1α, may contribute to mechanics by acting as physical linkers. Experimental data for nuclear mechanical response can only be reconciled with models which contain chromatin (an interior polymer), a lamina (a peripheral meshwork), and chromatin-chromatin and chromatin-lamina linkages (*Banigan et al., 2017*; *Hobson and Stephens, 2020b*; *Stephens et al., 2017*). Further studies have suggested that these linkages may govern nuclear shape stability (*Lionetti et al., 2020*; *Liu et al., 2021*; *Schreiner et al., 2015*). Experimental studies have shown chromatin linkages to the nuclear periphery aid shape stability and mechanics (*Schreiner et al., 2015*). Furthermore, recent chromosome conformation capture (Hi-C) and mechanics experiments suggest that chromatin is physically linked about once per 15 kb, since chromatin organization and mechanical response are perturbed only upon extreme chromatin fragmentation by restriction enzymes (*Belaghzal et al., 2021*). Whether chromatin-binding

proteins like HP1α provide mechanical and morphological stability to the nucleus and whether their function is to maintain histone modifications or act as physical linkers remains an open question.

Most studies of epigenetic modification of chromatin and nuclear mechanics have focused on the interphase nucleus. However, it is conceivable that some of the epigenetic marks involved in heterochromatin formation during interphase might survive and have effects during cell division. Consistent with this, recent work indicates that hypermethylation of histones can persist into metaphase and is correlated with increased stiffness of mitotic chromosomes/metaphase chromatin (*Biggs et al., 2019*). However, it remains unknown whether the readers of those marks, such as HP1α, contribute significantly to metaphase chromatin structure and mechanics and how important they are to ensuring the success of mitosis.

Here, we determine the mechanical role of heterochromatin protein HP1α and its independence from histone methylation. We created and characterized an auxin-inducible-degradation (AID, [*Nishimura et al., 2009*]) system for rapid depletion of endogenous HP1α in human U2OS cells. Using these novel CRISPR-derived HP1α-AID-sfGFP cells, we find that the transcriptional profile and chromatin organization are largely unchanged by rapid degradation of HP1α. However, rapid HP1α degradation causes decreased chromatin-based rigidity in both nuclei and mitotic chromosomes. Concurrently, we observe increases in aberrant nuclear morphology and incidence of mitotic errors, both of which are associated with disease. Increasing histone methylation rescues nuclear and mitotic chromosome mechanics associated with HP1α depletion, indicating that these factors contribute independently. Rescue experiments with a HP1α mutant protein reveal that its dimerization is essential for the maintenance of nuclear structure. Computational modeling supports the conclusion that HP1α's contribution to nuclear mechanics follows primarily from its function as a chromatin-chromatin crosslinker, suggesting that constitutive heterochromatin may be thought of as a polymer gel (*Colby et al., 1993*). These findings contribute to our understanding of the role of histone methylation and heterochromatin levels in controlling nuclear organization, mechanics, and morphology in healthy and diseased cell states.

## Results

### Rapid degradation of HP1α using an auxin-inducible degron

We generated a novel endogenous HP1α auxin-inducible degron for rapid and reversible depletion of HP1α protein in the cell. This was accomplished using CRISPR (*Doudna and Charpentier, 2014*) to tag both endogenous copies of the CBX5 gene in U2OS cells with an auxin-inducible degron (AID, [*Nishimura et al., 2009*]) and reporter Superfolder Green Fluorescent Protein (sfGFP) at the C terminus (HP1α-AID-sfGFP). Immunostaining demonstrated that modification of the endogenous loci did not alter the HP1α protein localization pattern (*Figure 1A*), while PCR, western blotting, and flow cytometry showed that all endogenous CBX5 alleles were tagged and only modified protein was expressed (*Figure 1C and D*, Materials and methods). Modification of the endogenous HP1α allele did not alter transcription or H3K9me[2,3] levels (see *Source data 3*, 76 upregulated and 56 downregulated transcripts, which represents a change in 0.8% of genes across >16,600; compare methylation levels in parental and tagged cells see *Figure 2—figure supplement 1D*). HP1α degradation was observed by fluorescence microscopy or flow cytometry of HP1α-AID-sfGFP cells and by western blot after 4 hr of treatment with 1 mM auxin (Indole-3-acetic acid, *Figure 1B-D*). These conditions consistently resulted in >90% degradation of HP1α. The degradation was reversible as protein levels recovered over 2 days after removal of auxin (*Figure 1B-D*). Thus, we report the novel generation of an endogenously tagged HP1α cell line, which has a fluorescent reporter and is capable of rapid, reversible degradation in hours.

Previous studies have shown that disruption of HP1α binding and localization through RNAi knockdown of its binding partners results in chromatin decompaction and loss of transcriptional silencing (*Frescas et al., 2008*; *Hahn et al., 2013*; *Shumaker et al., 2006*). Because tethering of HP1α to specific sites is sufficient to induce chromatin compaction and transcriptional silencing (*Li et al., 2003*; *Verschure et al., 2005*), we sought to determine whether rapid depletion of HP1α by auxin treatment would significantly alter global transcription or chromatin organization. RNA-Seq data was acquired, mapped (STAR), and quantified (RSEM), and the differential gene expression analysis was performed using DESeq2 for greater than 16,500 genes (see Materials and methods).

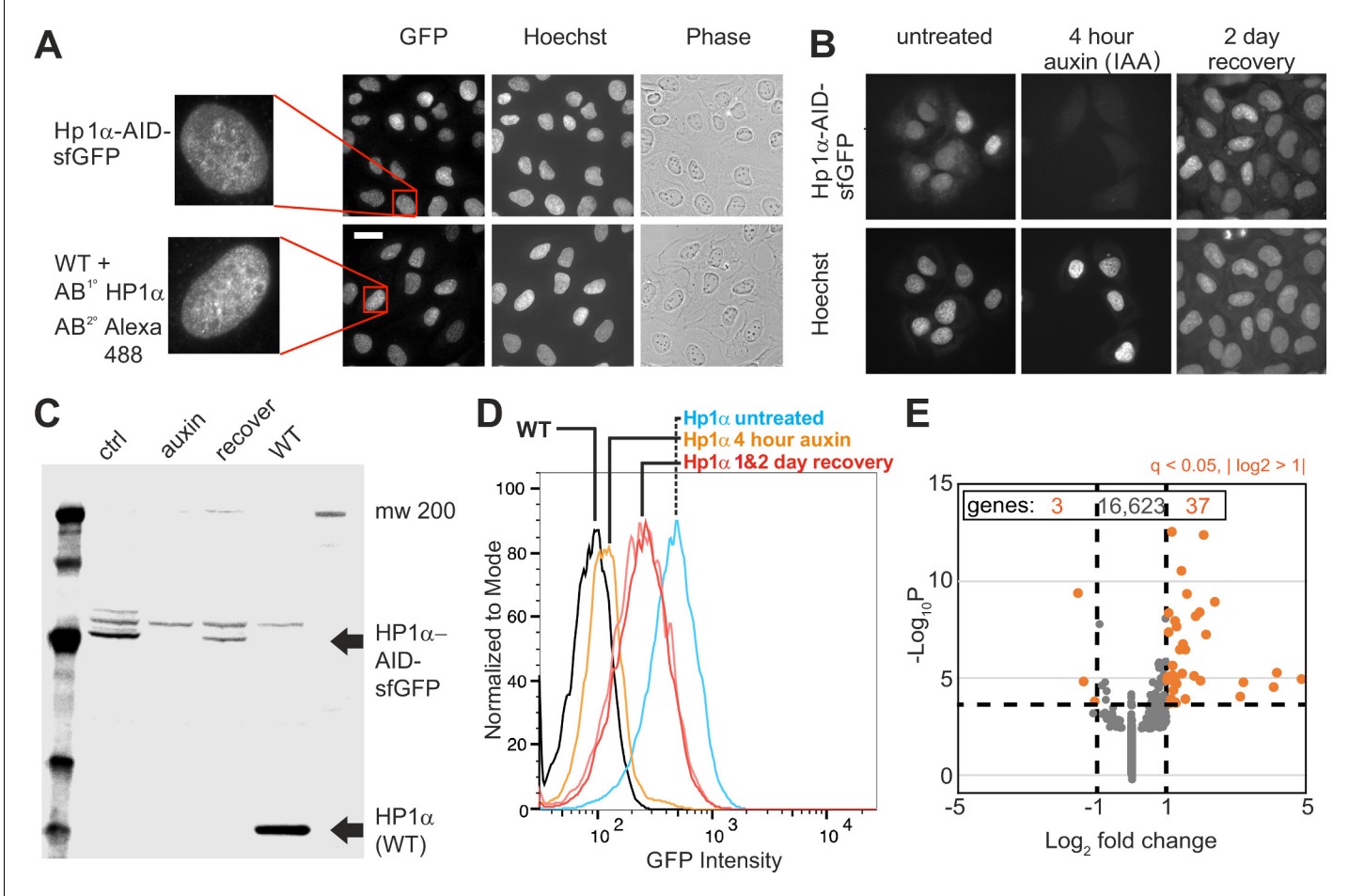

**Figure 1.** Generation of an CRISPR endogenous HP1α-auxin-inducible-degron-sfGFP cell line. (**A**) Example images of HP1α-AID-sfGFP relative to wild-type cells stained for HP1α via immunofluorescence along with Hoechst DNA stain and phase contrast images. Scale bar = 20 μm. (**B**) Example images of HP1α-AID-sfGFP before, after 4 hr of auxin treatment, and 2 days post auxin removal. Hoechst DNA stain aids labeling of nuclei. (**C**) Western blot and (**D**) Flow cytometer graph of GFP intensity of control (ctrl/untreated), auxin-treated for 4 hr, 2 days after removal of auxin, and wild-type (WT) showing short-term loss and recovery of HP1α-AID-sfGFP. (**E**) Graph of RNA-seq data showing that few genes change transcript levels as determined by q-value <0.05 (calculated via -Log$_{10}$P) and absolute change of Log$_2$ fold >1 (marked in orange), with expression of only 40 out of of 16,663 genes changing significantly comparing control/untreated versus 4 hr auxin-treated.

The online version of this article includes the following figure supplement(s) for figure 1:

**Figure supplement 1.** Local chromatin compaction remains after HP1α degradation.

Transcription analysis of HP1α-AID-sfGFP control and 4 hr auxin-treated cells revealed that only three genes were downregulated and only 37 genes were upregulated (q-value <0.05 and fold-change >2, *Figure 1E*, *Source data 1*). Lack of transcriptional changes upon rapid degradation of HP1α was further supported by comparing control and 16 hr of auxin treatment, which yielded 15 downregulated and four upregulated genes (see *Source data 2*). These data are similar to previous reports that in mammalian systems, HP1 proteins are not required for maintenance of silencing (*Maksakova et al., 2011*). Furthermore, satellite derepression and other transcriptional changes previously reported after redistribution of HP1α may be unique to each organism and be indirect or dependent on secondary chromatin rearrangements. In addition, DAPI and Hoechst staining patterns showed similar dense regions of nuclear stain typical of heterochromatin in both treated and untreated cells (*Figure 1* and *Figure 1—figure supplement 1*). Furthermore, histone density and distribution do not significantly change on a single-cell basis (*Figure 1—figure supplement 1*). These results indicate that no global change in transcription or global and local changes in histone density occurred after rapid HP1α degradation.

## HP1α is a major mechanical component of the interphase nucleus that contributes to nuclear shape maintenance

We hypothesized that HP1α could aid nuclear mechanics due to its association with heterochromatin. To test this hypothesis, we perform single-nucleus micromanipulation force measurements on untreated and auxin-induced HP1α-degraded nuclei. Micromanipulation is an extensional force measurement technique capable of separating chromatin- and lamin-based nuclear mechanics (*Stephens et al., 2017*). First, a single nucleus is isolated from a living cell following treatment with latrunculin A to depolymerize actin and local lysis applied via micropipette spray (*Figure 2A*). The isolated nucleus is then loaded between two micropipettes. One micropipette is moved to extend the nucleus, while the other micropipette's deflection, multiplied by the premeasured bending constant, measures force (*Figure 2A*). The force-extension relation is nonlinear, but can be decomposed into two linear slopes, which provide nuclear spring constants (nN/μm) for the short-extension regime (<3 μm), quantifying chromatin-based stiffness, and the long-extension regime (>3 μm), quantifying lamin-based strain stiffening (*Stephens et al., 2019b*; *Stephens et al., 2018*; *Stephens et al., 2017*; example *Figure 2B*).

Micromanipulation force measurements reveal that degradation of HP1α affects nuclear mechanics. Parental unmodified U2OS cells, control or auxin-treated, show no change in the chromatin-based nuclear spring constant (0.35 vs. 0.34 ± 0.06 nN/μm, p=0.73, *Figure 2C*), and nuclear mechanics of cells with tagged HP1α (vs. 0.40 ± 0.03 nN/μm, p=0.42, *Figure 2D*) are not significantly different from parental control. This suggests that auxin treatment alone does not alter nuclear mechanics, and the addition of the AID-sfGFP tag to HP1α does not alter normal nuclear mechanical response. HP1α-AID-sfGFP cells were imaged before nucleus isolation to verify presence or absence of HP1α via sfGFP reporter. Auxin-induced HP1α degradation resulted in a 45% decrease in short-extension chromatin-based nuclear stiffness (0.40 vs 0.22 ± 0.03 nN/μm, p<0.001, *Figure 2D*). However, long-extension strain stiffening remained relatively unchanged (example, *Figure 2B*; *Figure 2—figure supplement 1A*, p=0.99) in agreement with the observation of no decrease in lamin A/C or B1 levels (*Figure 2—figure supplement 1, B C*). This data indicates that HP1α contributes to chromatin mechanics of the cell nucleus.

Previous work has shown that nuclear softening due to perturbations of chromatin and its mechanics, particularly the loss of heterochromatin, can induce abnormal nuclear morphology (*Stephens et al., 2019b*; *Stephens et al., 2018*). Consistent with these prior findings, we quantified nuclear shape by shape solidity (ratio of area to convex area) and found that HP1α-degraded cells displayed a statistically significant decrease in average solidity (0.971 control vs. 0.969 auxin, p=0.005). Strikingly, we observe a large increase in the fraction of nuclei that have low levels of solidity, which we refer to as abnormal nuclei. Abnormal nuclei increase from 10 ± 1% in untreated cells to 22 ± 5% upon HP1α loss, as quantified by nuclei below a specified solidity threshold (solidity <0.96, *Figure 2G*). Another way to quantify shape is average nuclear curvature, which increases when the nucleus deviates from its normal elliptical shape (see Materials and methods). Tracking average nuclear curvature of single nuclei over time post auxin treatment reveals a significant increase in nuclear curvature during interphase (0% control vs 36% HP1α-degraded single nuclei persistently exhibit curvature increased by >0.05 μm$^{-1}$, *Figure 2—figure supplement 2*), coincident with HP1α loss and decreased nuclear stiffness (4 hr auxin, *Figure 2*). Loss of nuclear mechanics and shape has been shown to cause dysfunction via nuclear ruptures and increased DNA damage (*Stephens et al., 2019b*; *Stephens et al., 2018*; *Xia et al., 2018*). We found similar results upon degradation of HP1α, as we observed dispersal of NLS-RFP into the cytoplasm during loss of nuclear compartmentalization by ruptures and a doubling of DNA damage as measured by ϒH2AX foci (*Figure 2—figure supplement 3*). These results agree with siRNA knockdown of HP1α in U2OS cells, which have demonstrated accumulation of DNA damage foci in a previous report (*Lee et al., 2013*). These data establish that HP1α degradation results in nuclear softening, abnormal nuclear morphology, and nuclear dysfunction.

## HP1α and histone methylation contribute independently to nuclear mechanics and morphology

It is unclear exactly how the different components of heterochromatin work together to define its structure and function and how dependent they are on one another. For example, studies using

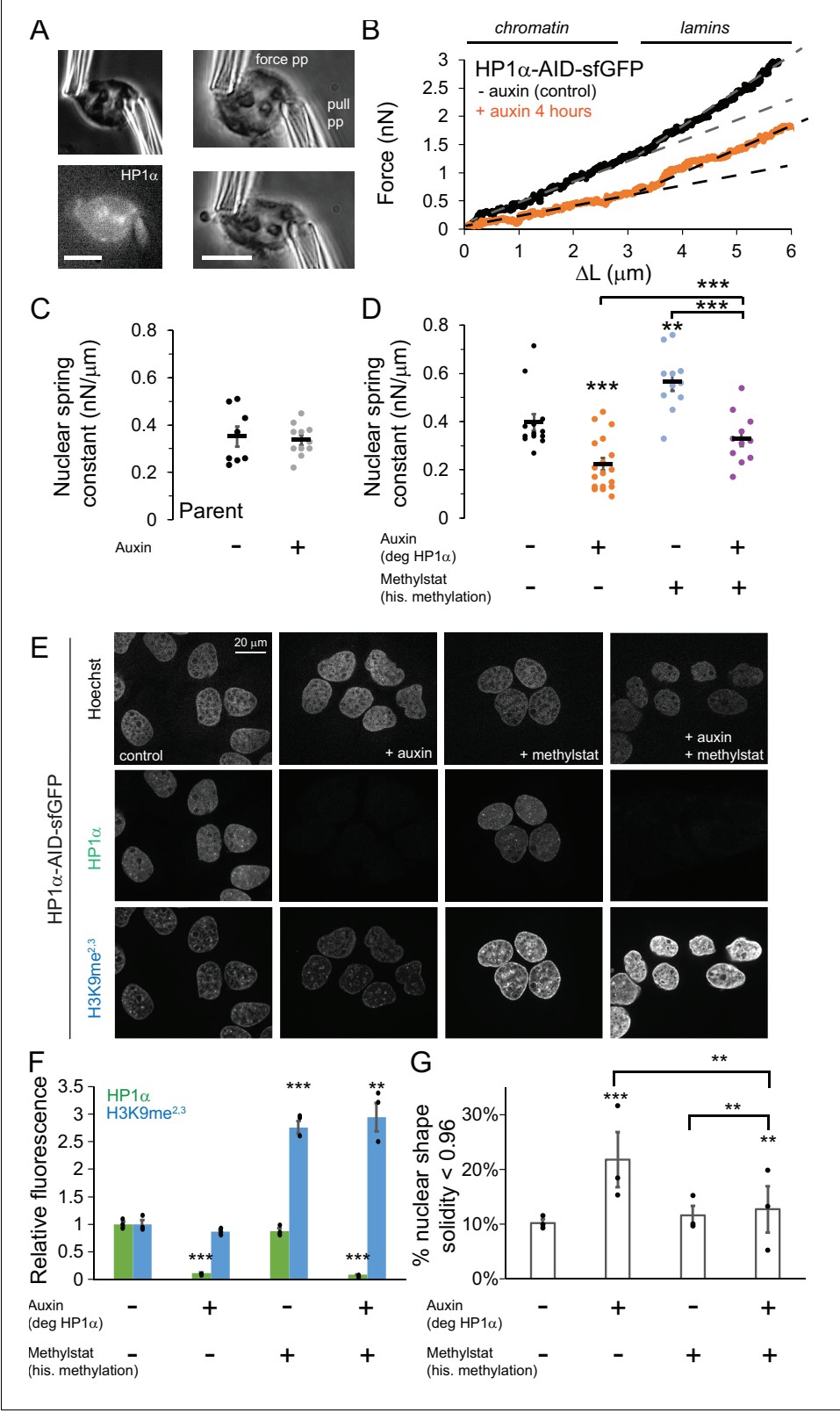

**Figure 2.** HP1α is a mechanical component of the nucleus controlling nuclear shape, separately from histone methylation. (**A**) Example images of a single isolated nucleus via transmitted light and HP1α-AID-sfGFP fluorescence and single nucleus micromanipulation force-extension measurement experiment. The pull pipette extends the nucleus while the bending of a premeasured force pipette provides the force measurement. Scale bar = 10 μm. (**B**) Example traces of micromanipulation force-extension for control (black) and auxin-induced degradation of HP1α (orange) provide a measure of

*Figure 2 continued on next page*

*Figure 2 continued*

nuclear spring constant from the slope (dotted lines). Initial slope provides chromatin-based nuclear spring constant while the second slope provides the lamin-based strain stiffening nuclear spring constant. (**C and D**) Graphs of average and single chromatin-based nuclear spring constant for (**C**) parental cell line control and 4–6 hr auxin treated and (**D**) HP1α-AID-sfGFP with and without auxin and/or methylstat treatment. n = 11–18 nuclei each. (**E**) Example images of cells treated with and without auxin and/or methylstat. (**F**) Quantified relative fluorescence of HP1α-AID-sfGFP and heterochromatin marker H3K9me[2,3]. (**G**) Quantified abnormal nuclear morphology determined as solidity value less than 0.96, statistics via chi-squared analysis. Another way to quantify abnormal nuclear morphology is via average nuclear curvature reported in *Figure 2—figure supplement 2*. Three biological experiments (shown as black dots) each consisting of n = 109, 102, 105 control; n = 137, 115, 165 auxin, n = 31, 34, 32 methylstat, and n = 102, 92, 78 auxin methylstat. Average measurements were similar for control, methystat, and auxin with methylstat 0.971 ± 0.0001 but different for auxin 0.969 ± 0.0015, p=0.005. p values reported as *<0.05, **<0.01, ***<0.001, no asterisk denotes no significance, p>0.05. Error bars represent standard error.

The online version of this article includes the following figure supplement(s) for figure 2:

**Figure supplement 1.** Lamin mechanics and levels do not change upon degradation of HP1α.

**Figure supplement 2.** Increased nuclear curvature onsets coincident with HP1α loss during interphase.

**Figure supplement 3.** HP1α-degraded nuclei display nuclear rupture, increased DNA damage, and abnormal nuclear morphology post mitosis.

genetic knockouts (*Bosch-Presegué et al., 2017*) and long-term depletion RNAi studies of heterochromatic components (*Frescas et al., 2008*; *Hahn et al., 2013*; *Shumaker et al., 2006*) have reported that HP1α not only binds to methylated histones, but also aids in histone methylation establishment and maintenance (*Jacobs and Khorasanizadeh, 2002*; *Nielsen et al., 2002*; *Schotta et al., 2002*). HP1α could simply alter levels of H3K9-methylated histones to affect nuclear mechanics and morphology. Levels of the constitutive heterochromatin mark H3K9me[2,3] did not change significantly after 16 hr or 96 hr of HP1α depletion (*Figure 2*, E and F; *Figure 2—figure supplement 2, D E* with H3K9ac). Thus, while rapid reduction of HP1α levels affects nuclear mechanics and morphology, it does not cause significant changes in histone methylation.

Previous reports have shown that increased histone methylation stiffens the nucleus (*Stephens et al., 2019b*; *Stephens et al., 2018*; *Stephens et al., 2017*). Cells were treated with the broad histone demethylase inhibitor methylstat, which increases H3K9 methylation approximately three-fold over its normal levels in the HP1α-AID-sfGFP cell line (*Figure 2*, E and F). Micromanipulation force experiments with HP1α-AID-sfGFP cells treated with 1 μM methylstat for 48 hr measured a stiffer chromatin-based nuclear spring constant (control 0.40 vs. methylstat 0.56 ± 0.03 nN/μm, p=0.003, *Figure 2D*), similar to previously reported experiments on different cell lines (*Stephens et al., 2018*). Increased broad histone methylation via methylstat did not significantly increase HP1α-AID-sfGFP levels (*Figure 2*, E and F). Thus, chromatin-based nuclear mechanics can be modulated by changing either HP1α levels or methylated histone levels.

We reasoned that elevating levels of methylated histone in HP1α-degraded nuclei would reveal the relative contributions of histone methylation and HP1α to nuclear mechanics and shape. If chromatin mechanics is dictated entirely by HP1α, increasing histone methylation in auxin-treated cells should not change nuclear mechanics; in that case, the nuclear spring constant should match that of the HP1α-degraded cells. Alternatively, if the methylation state of histones contributes to chromatin stiffness independently of HP1α, methylstat-treated HP1α-degraded nuclei will have a larger spring constant than HP1α-degraded nuclei and may display rescued nuclear shape.

Experiments are consistent with the second scenario, where increasing histone methylation levels in HP1α-degraded cells resulted in rescued nuclear mechanics and shape. Micromanipulation force measurements reveal a larger nuclear spring constant for HP1α-degraded nuclei with increased histone methylation as compared to HP1α-degraded with normal levels of methylation, returning to a spring constant similar to wild-type levels (auxin 0.22 vs. auxin+methylstat 0.33 ± 0.03 nN/μm, p<0.001, *Figure 2D*). Alternatively, compared to normal levels of HP1α with increased histone methylation, loss of HP1α and increased histone methylation resulted in a decreased nuclear spring constant (auxin+methylstat 0.33 vs. methylstat 0.56 ± 0.03 nN/μm, p<0.001, *Figure 2D*). Strain stiffening in the lamin-dependent regime remained similar across all treatments (*Figure 2—figure supplement 1*). Consistent with the mechanical measurements, methylstat treatment rescues abnormal morphology associated with HP1α degradation from 22% abnormal to 13% (*Figure 2G*). Altogether, these results suggest that HP1α and methylated histone levels both contribute to chromatin-based nuclear mechanics and morphology. Moreover, the approximately additive nature of the

changes in nuclear stiffness, along with the lack of interdependence between levels HP1α and histone methylation, suggest that these mechanisms contribute to mechanics independently.

## Maintenance of nuclear morphology depends on HP1α dimerization

HP1α forms a homodimer that can bridge strands of chromatin by binding two H3K9me[2,3] marks on different nucleosomes through its chromodomain (*Jacobs and Khorasanizadeh, 2002*; *Machida et al., 2018*; *Nielsen et al., 2002*) or two strands of DNA through a positively charged KRK patch in the hinge (*Larson et al., 2017*). We reasoned that the role of HP1α in determining nuclear shape and mechanics, independent of histone methylation levels, might be due to its ability to physically crosslink chromatin strands. This linking ability would be dependent on HP1α dimerization, which can be disrupted with a point mutant, HP1α[I165E] (*Brasher et al., 2000*; *Lechner et al., 2005*; *Lechner et al., 2000*; *Thiru et al., 2004*). To determine if dimerization is key to its mechanical and morphological contributions in vivo, we asked whether a non-dimerizing mutant (HP1α[I165E]) could rescue nuclear morphology when the endogenous protein was degraded.

HP1α-AID-sfGFP cells were infected with lentivirus to stably express either exogenous HP1α[WT]-mCherry (positive control) or HP1α[I165E]-mCherry (dimer mutant, [*Brasher et al., 2000*]) under an SFFV promoter. Two days post-infection, these cells stably expressing HP1α[WT]-mCherry or HP1α[I165E]-mCherry were treated with 1 mM auxin for 16 hr to degrade the endogenous HP1α-AID-sfGFP and assess the ability of the rescue construct to maintain and recover normal nuclear shape. These cells were fixed and immunostained for lamin A/C, and then the shape of the nucleus was quantified using average curvature of each nucleus (see Materials and methods).

We first measured nuclear curvature in parental U2OS, and control and auxin-treated HP1α-AID-sfGFP cells to determine normal and abnormal nuclear curvature, respectively. HP1α-AID-sfGFP cells have no difference in nuclear shape compared to parental U2OS cells (parental $0.114 \pm 0.002$ μm$^{-1}$; HP1α-AID-sfGFP $0.118 \pm 0.001$ μm$^{-1}$, p>0.05), and auxin-treatment of parental cells does not alter their nuclear morphology (parental - auxin $0.114 \pm 0.002$ μm$^{-1}$; parental + auxin $0.117 \pm 0.001$ μm$^{-1}$, p>0.05). Similar to our previous measurements above (*Figure 2* and *Figure 2—figure supplement 2*), auxin-treated HP1α-degraded nuclei exhibited higher average nuclear curvature compared

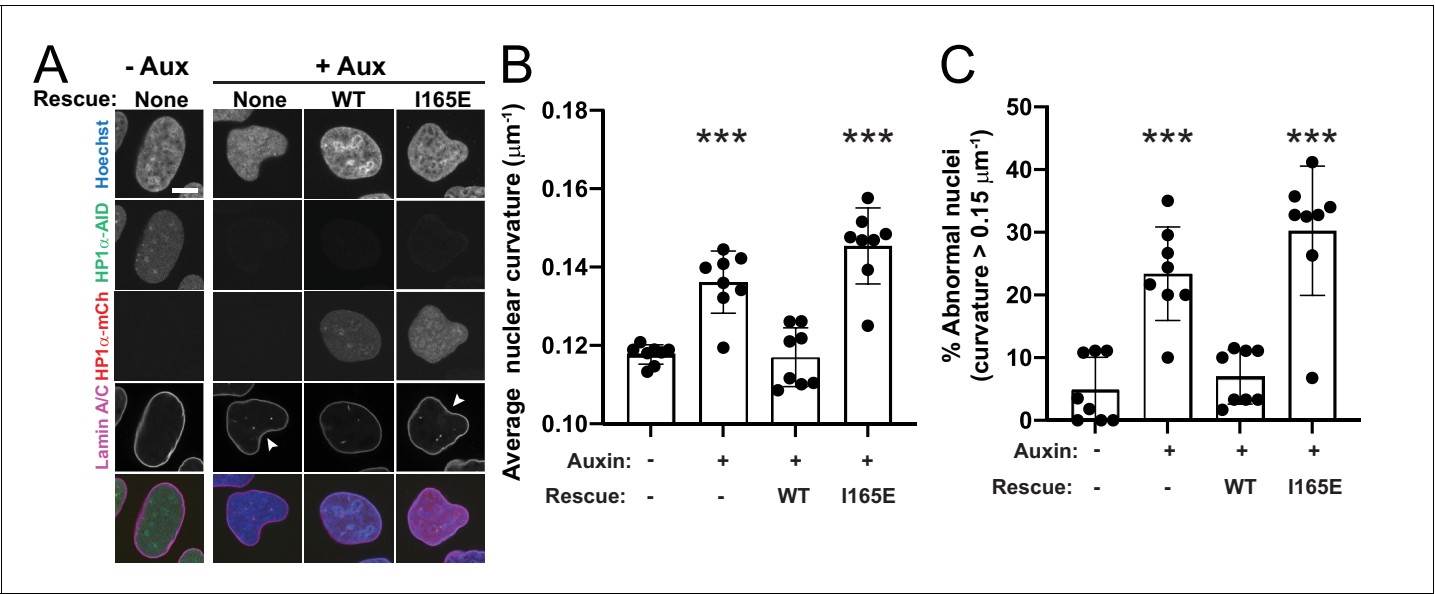

**Figure 3.** HP1α dimerization is essential for maintenance of nuclear shape. (**A**) Example images of HP1α-AID-sfGFP cells control (-Aux) and auxin treated (+Aux) with and without exogenous HP1α wild-type (WT) or dimer mutant (I165E) rescue constructs tagged with mCherry. Scale bar = 10 μm. (**B**) Graph of average nuclear curvature measurements with individual trials as black dots. (**C**) Graph of percentage of abnormally shaped nuclei, determined as greater than 0.15 μm$^{-1}$ curvature, which is the average untreated nucleus plus the standard deviation. Eight experimental biological replicates were measured for each condition (denoted as black dots) consisting of -auxin, n = 37, 45, 45, 57, 58, 57, 55, 54; +auxin, n = 60, 44, 41, 60, 60, 60, 60, 60; +auxin and WT exogenous rescue, n = 27, 30, 36, 60, 60, 60, 60, 19; +auxin and I165E exogenous rescue, n = 38, 40, 50, 59, 56, 58, 58, 51. p values reported as no asterisk >0.05, *<0.05, **<0.01, ***<0.001, calculated by one-way ANOVA. Error bars represent standard error.

to control HP1α-AID-sfGFP nuclei (*Figure 3*, control 0.116 ± 0.003 µm$^{-1}$ vs. HP1α-degraded with no rescue, 0.142 ± 0.001 µm$^{-1}$, p=0.0002). Expression of exogenous HP1α$^{WT}$-mCherry in auxin-treated HP1α-AID-sfGFP-degraded cells rescued nuclear morphology to near wild-type levels (p=0.99, control vs. HP1α-degraded with HP1α$^{WT}$ rescue, 0.123 ± 0.002 µm$^{-1}$). However, the dimer mutant HP1α$^{I165E}$-mCherry did not rescue nuclear morphology in auxin-treated cells (*Figure 3*, control vs. HP1α-degraded with HP1α$^{I165E}$ rescue, 0.151 ± 0.003 µm$^{-1}$, p<0.0001). We observed that a subset of nuclei had abnormal nuclear shape (curvature greater than one standard deviation above control). Using this metric, cells in which HP1α-AID-sfGFP was degraded displayed a higher level of abnormally shaped nuclei compared to control (30% vs. 11%, *Figure 3C*). Expression of HP1α$^{WT}$-mCherry recovered WT levels (13% abnormal), while expression of HP1α$^{I165E}$-mCherry did not, leaving many abnormally shaped nuclei (31%). Together, these results indicate that HP1α dimerization is essential to its function in nuclear morphology and indicates that bridging or crosslinking of chromatin fibers is important in determining nuclear shape.

## Simulations of nuclear mechanics modulating chromatin crosslinking recapitulate experimental degradation of HP1α

To assess the role of HP1α in chromatin-based nuclear mechanical response, we performed Brownian dynamics simulations using a previously developed shell-polymer model (*Banigan, 2021*; *Banigan et al., 2017*; *Stephens et al., 2017*). In these simulations, chromatin is modeled as a crosslinked polymer that is physically linked to a peripheral polymeric lamin shell that encapsulates the polymer chromatin (see Materials and methods). In this model, each chromatin bead is 0.57 µm in diameter and represents a few Mbp of the genome. This coarse-grained model can capture the effects of alterations to histone modifications through the polymer spring constant and perturbations to lamin A/C through the lamin spring constant (light red data points in *Figure 4*, A and C; *Stephens et al., 2017*). In particular, varying the polymer spring constant models alterations to chromatin compaction via histone modifications; the short-extension nuclear force response is suppressed as the polymer spring constant is decreased (*Stephens et al., 2017*).

However, rapid depletion of HP1α does not alter histone methylation state or lamin expression levels (*Figure 2*, E and F; *Figure 2—figure supplement 2*), so we sought to identify a distinct physical role for HP1α within this framework. We hypothesized that HP1α might instead govern mechanics either by linking heterochromatin to the lamina via proteins such as PRR14 (*Poleshko et al., 2013*) and LBR (*Polioudaki et al., 2001*; *Ye et al., 1997*), or by binding and bridging nucleosomes (*Azzaz et al., 2014*; *Canzio et al., 2011*; *Erdel et al., 2020*; *Machida et al., 2018*). Thus, we explored whether HP1α might impact nuclear mechanical response by forming chromatin-chromatin crosslinks or by forming chromatin-lamina linkages.

We first investigated whether depletion of chromatin-lamina linkages in the simulations could generate the same mechanical effects as HP1α degradation in the experiments. In simulations, we varied the frequency of linkages between the chromatin and the lamina from zero up to ~50% of the chromatin subunits that reside near the shell. We found that the frequency of chromatin-lamina linkages affects the two-regime force response of the model nucleus (*Figure 4A*). The spring constants quantifying both the short- and long-extension force responses decrease as the number of chromatin-lamina linkages is decreased (*Figure 4B*). With fewer chromatin-lamina linkages, the mechanical coupling between the nuclear periphery and the interior is lost, which suppresses short-extension rigidity; simultaneously, the loss of these linkages uncouples the lamina from the stiff chromatin interior, which also decreases the long-extension stiffness. This result contrasts with measurements from the micromanipulation experiments (*Figure 2D*; *Figure 2—figure supplement 1*), which show that the short-extension spring constant, but not the long-extension spring constant, decreases after HP1α degradation. Thus, we conclude that the mechanical contributions of HP1α more likely arise from an alternative structural function.

We therefore investigated the effects of varying the levels of chromatin-chromatin crosslinkers in the simulation model. We varied crosslinking frequency from zero up to about one in three subunits crosslinked, above which the chromatin polymer is a percolated network and therefore solid-like. We found that the level of crosslinking markedly alters the force-strain relation (*Figure 4C*); increasing crosslinking stiffens the nucleus. However, in contrast to chromatin-lamina linkages, crosslinks govern stiffness of only the short-extension force response (*Figure 4D*). This is a signature of their specific effect in resisting deformations of the chromatin interior. These qualitative trends agree with

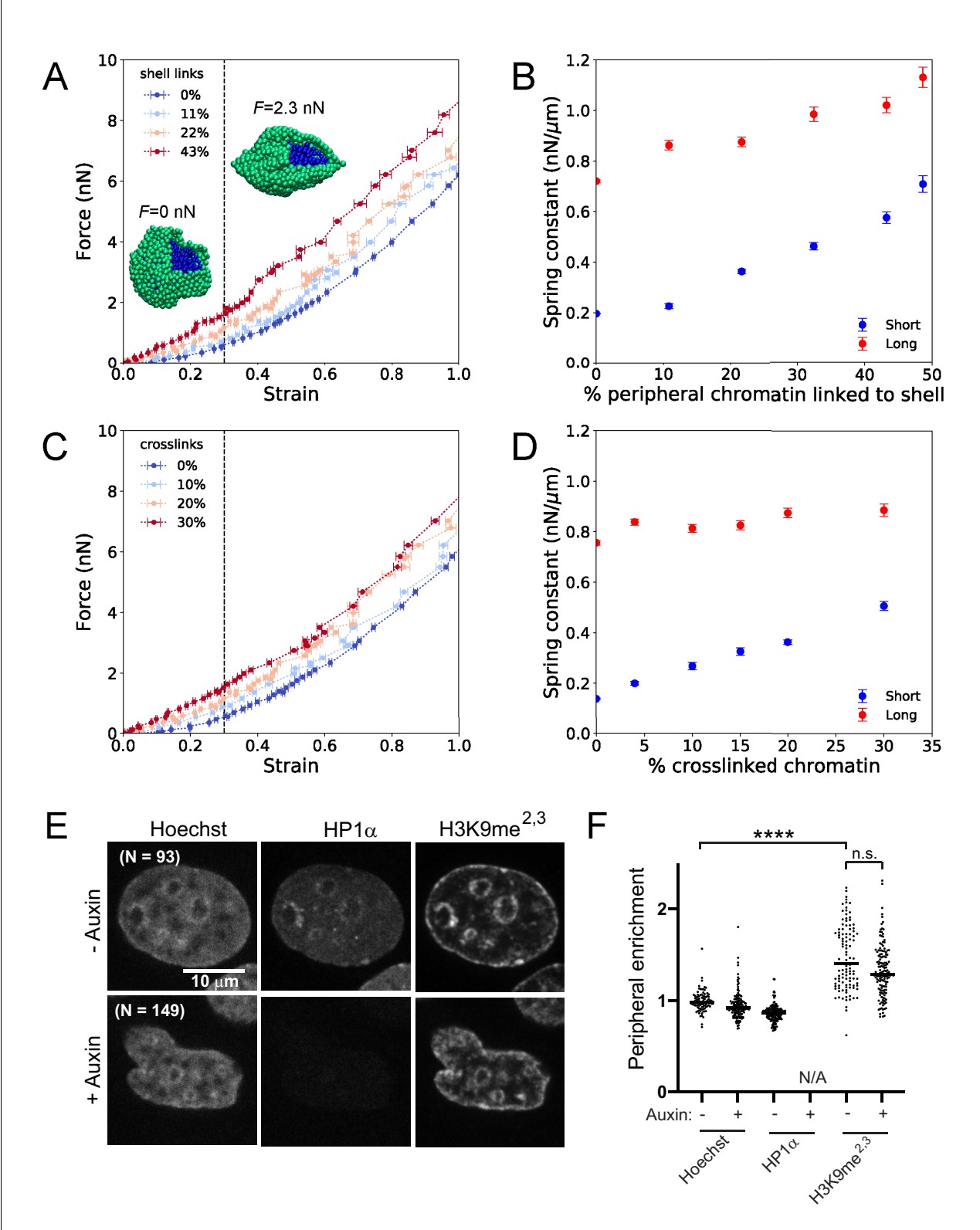

**Figure 4.** Simulations of nuclear mechanical response and experimental measurements of peripheral heterochromatin support a model with HP1α as a chromatin-chromatin crosslinker. (**A**) Force-strain relationship for simulated nuclei with various levels of chromatin-lamina (shell) linkages. Colors indicate different percentages of chromatin segments linked to the lamina. *Insets*: Snapshots of simulations with a portion of the lamina (green) removed to reveal the interior chromatin (blue) for two different applied stretching forces, *F*. (**B**) Spring constants for short- and long-extension regimes for

*Figure 4 continued on next page*

*Figure 4 continued*

simulations with various levels of chromatin-lamina crosslinks, quantified by percentage of peripheral chromatin subunits linked to the shell (blue and red, respectively). (C) Force-strain relation for simulated nuclei with various levels of chromatin-chromatin crosslinks. (D) Spring constants for short- and long-extension with varying levels of chromatin-chromatin crosslinks (blue and red). Vertical dashed lines in (A) and (C) separate the short-extension and long-extension regimes. Each force-strain data point in (A) and (C) is an average that is computed from n ≥ 11 simulations. Short-extension spring constants in (B) and (D) are each computed from $n_{short} \geq 13$ and 10 force-extension data points, respectively. Long-extension spring constants in (B) and (D) are each computed from $n_{long} \geq 19$ and 15 force-extension data points, respectively. (E) Example images of HP1α-AID-sfGFP nuclei untreated or auxin-treated, analyzed for (F) enrichment measurements (peripheral/internal average signal) to determine peripheral enrichment of DNA (Hoechst), HP1α, and H3K9me[2,3]. p values denoted as n.s. >0.05 and ****<0.0001, calculated by one-way ANOVA. Error bars in (A)-(D) show standard error of the mean.

the measurements from micromanipulation experiments (*Figure 2*, B and D). The simulation data also includes points that are in reasonable quantitative agreement with the experiments. These results are consistent with a model in which the HP1α[I165E] mutant abolishes crosslinking and thus decreases the short-extension nuclear spring constant (*Figure 2*, B and D), which may generate abnormal nuclear morphology. Altogether, the simulations support the conclusion that HP1α contributes to nuclear mechanical response by acting as a chromatin-chromatin crosslinking element.

## HP1α degradation does not release heterochromatin from the nuclear periphery

To test our prediction from simulations that HP1α does not act mechanically as a chromatin-lamina linker, we experimentally assayed its location and peripheral heterochromatin tethering capabilities. Specifically, we investigated whether HP1α acts similarly to two known chromatin-lamina tethers, LBR and PRR14, which show enrichment at the periphery and maintain localization of peripheral H3K9-marked heterochromatin (*Dunlevy et al., 2020*; *Giannios et al., 2017*; *Nikolakaki et al., 2017*; *Poleshko et al., 2013*; *Solovei et al., 2013*). We measured peripheral enrichment ratios (average intensity at the periphery over the interior) of DNA (Hoechst), HP1α, and H3K9me[2,3], using lamin B1 as a marker for the periphery. In untreated cells, both DNA and HP1α have enrichment ratios of about 1 (0.99 ± 0.01 vs. HP1α 0.86 ± 0.01, p=0.78, *Figure 4*, E and F), demonstrating a lack of peripheral enrichment, while H3K9me[2,3] was somewhat enriched (1.48 ± 0.04, p<0.0001 vs. 0.99 DNA). Upon degradation of HP1α, peripheral enrichment of DNA and H3K9me[2,3] do not change (0.99 vs 0.94, p=0.99 and 1.48 vs 1.34, respectively, p=0.49), whereas depletion of previously reported chromatin-lamina tethers result in a 50% or greater decrease in peripheral localization of H3K9me[2,3] (*Poleshko et al., 2013*), which supports our conclusion that HP1α does not mechanically function as a chromatin-lamina linkage in this cell type.

## HP1α provides mechanical strength to mitotic chromosomes and enhances mitotic fidelity

Given HP1α's mechanical role in chromatin-based nuclear mechanics, we hypothesized that HP1α could also contribute to mitotic chromosome mechanics. As in interphase nuclear mechanical response, heterochromatin has recently been shown to govern mitotic chromosome mechanics (*Biggs et al., 2019*). It has previously been reported that most HP1α is removed from chromosomes during prophase by phosphorylation of H3S10, which is known to disrupt HP1α-H3K9me[2,3] binding (*Fischle et al., 2005*; *Hirota et al., 2005*). However, some HP1α binding is maintained throughout mitosis (*Serrano et al., 2009*), suggesting a possible role for HP1α in mitotic chromosome mechanics.

We used fluorescence imaging and micropipette micromanipulation methods (*Biggs et al., 2019*; *Sun et al., 2018*) to assay the presence of HP1α-AID-sfGFP in prometaphase cells (identified by their round shape) and mitotic chromosomes without or with auxin treatment for 4 hr to degrade HP1α (*Figure 5B*). Prometaphase cells show both chromosome-bound and diffuse, cytoplasmic HP1α-AID-sfGFP signals. Both cytoplasmic and chromosomal HP1α-AID-sfGFP signals nearly completely disappear upon auxin-induced degradation (*Figure 5B*). To further verify the presence of HP1α on mitotic chromosomes, we isolated mitotic chromosome bundles from cells via gentle lysis and capture. Fluorescence imaging of these isolated bundles without the high background fluorescence of the cytoplasm allowed us to observe that HP1α is clearly present on mitotic chromosomes (*Figure 5A*). In

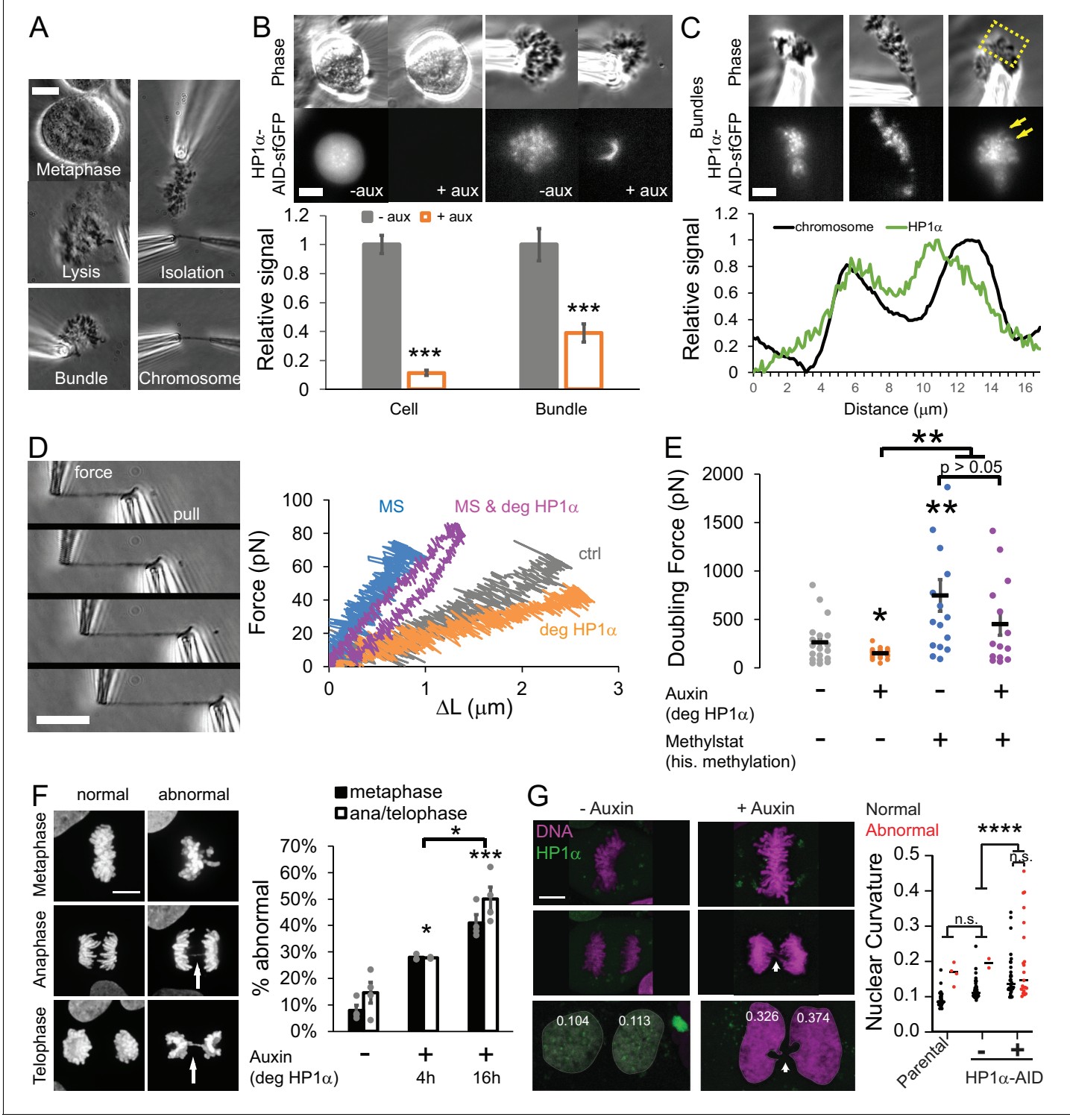

**Figure 5.** HP1α is a mechanical component of the mitotic chromosome aiding proper segregation in mitosis. (**A**) Example image of the steps to isolating a mitotic chromosome from a live cell using micropipettes. (**B**) Representative live mitotic cells and isolated mitotic chromosome bundles imaged via phase contrast and HP1α-AID-sfGFP fluorescence intensity across treatments. Values calculated by measuring the cell's or chromosome bundle's fluorescence minus the background fluorescence, normalized to the average intensity of the untreated cellular HP1α fluorescent intensity. p Values reported as ***<0.001, calculated by student's t-test. (**C**) Example images of the endogenous HP1α-AID-sfGFP fluorescence of an isolated mitotic bundle outside of the lysed cell. Yellow box denotes the area where the graphed line scan was drawn. The line scan reveals HP1α on chromosome arms. (**D**) Example images of a force-extension experiment. The right pipette pulls away from the left pipette, which stretches the

*Figure 5 continued on next page*

*Figure 5 continued*

chromosome and causes the left pipette to deflect. The left 'force' pipette has a premeasured bending constant (in pN/um) to calculate force. Left graph, example traces of force-extension experiments for the different conditions. (E) Graph of average doubling force (100% strain) in picoNewtons for each condition, which is determined by slope of the force extension traces and the initial chromosomes length. For B-E, n = 20 for control and auxin treated, n = 16 for methylstat, and n = 14 auxin methylstat treated, p values calculated by student's t-test. (F) Example images of abnormal mitotic segregation via anaphase bridge or nondisjunction. Graphs of percentage of mitotic cells displaying abnormal metaphase misalignment (black bars) and anaphase/telophase missegregation (white bars) via presence of anaphase bridges or nondisjunction/aneuploidy in control untreated cells (-) or auxin-treated (+) cells for 4 or 16 hr. Metaphase misalignment three to four biological replicate experiments (black dots) consisting of n = 16, 15, 20, 37 -aux, n = 33, 33, 24 +aux 4 hr, n = 22, 48, 58, 54 +aux 16 hr. Anaphase and telophase missegregation 3–4 experiments (black dots) consisting of n = 29, 23, 30, 30 -aux, n = 32, 29, 18 +aux 4 hr, n = 20, 35, 36, 45 +aux 16 hr. p values reported as *<0.05, **<0.01, ***<0.001, ****<0.0001, calculated by Student's t-test. (G) HP1α-AID-sfGFP cells - auxin or +auxin for 24 hr were tracked through mitosis to determine if abnormal mitosis results in abnormally shaped daughter nuclei measured via nuclear curvature (parental 34 nuclei from 17 mitoses; -auxin 46 nuclei from 23 mitoses; +auxin 51 nuclei from 26 mitoses, p value from one-way ANOVA). Percentage of abnormal mitosis presented in *Figure 5—figure supplement 1D*. Error bars represent standard error. Scale bar in A-C = 10 μm and F = 20 μm.

The online version of this article includes the following figure supplement(s) for figure 5:

**Figure supplement 1.** HP1α-AID-sfGFP displays cell and chromosome fluorescence and upon its degradation mitotic errors occur leading to higher curvature nuclei.

addition to concentrated foci, HP1α-AID-sfGFP is also present on chromosome arms (*Figure 5C* and *Figure 5—figure supplement 1, A B*). Confocal imaging of live cells revealed that concentrated foci are located at the pericentromeric region (*Figure 5—figure supplement 1C*), in agreement with previously published reports (*Akram et al., 2018*; *Fischle et al., 2005*; *Hirota et al., 2005*; *Serrano et al., 2009*). By additional fluorescence imaging, we observed that HP1α-AID-sfGFP is lost upon auxin-induced degradation (*Figure 5B C*). Thus, we confirmed that endogenous HP1α-AID-sfGFP is associated with mitotic chromosome arms and pericentromeres, and it is degraded after 4 hr of auxin treatment.

The mechanical role of HP1α in mitotic chromosomes was investigated by micromanipulation force measurements. The isolated bundle of chromosomes was held by one micropipette while two additional micropipettes were used to capture and isolate a single chromosome (*Figure 5D*). The single mitotic chromosome is then extended with the stiff pull pipette, while deflection of the other, much less stiff force pipette provides a force measurement, in the same manner as our experiments on interphase nuclei (*Figure 5D*). For each isolated chromosome, we calculated a force versus extension plot (*Figure 5D*). Because each of the 23 human chromosomes is a unique length, we calculate a length-independent measurement by extrapolating the force-extension slope to determine the 'doubling force'—the force at which the chromosome length would be doubled (i.e. force at 100% strain, *Figure 5E*). Since the pipettes hold opposite ends of the chromosome, tension is distributed across the whole chromosome (*Figure 5D* example images). Therefore, the resistive force measured includes contributions from chromatin, and thus HP1α, in both the chromosome arms and the pericentromeric region. We find that depletion of HP1α reduced mitotic chromosome doubling force by approximately 40%, from 262 ± 50 pN in control cells (spring constant 27 pN/μm) to 148 ± 12 pN in auxin-treated cells (16 pN/μm) (p=0.03, *Figure 5E*), indicating that HP1α significantly contributes to mitotic chromosome mechanics.

We next investigated whether histone methylation and the HP1α protein separately govern chromosome mechanics during mitosis, as they do during interphase. Increasing histone methylation via methylstat treatment has previously been shown to play a critical role in mechanical stiffness of mitotic chromosomes (*Biggs et al., 2019*). Furthermore, evidence exists for direct biochemical interactions between epigenetic marks on nucleosomes, independent of mark-reading proteins such as HP1α (*Bilokapic et al., 2018*; *Zhiteneva et al., 2017*). Thus, we aimed to determine whether histone methylation and HP1α contribute independently to mitotic chromosome stiffness.

We treated cells with the histone demethylase inhibitor methylstat to increase levels of methylated histones in cells with or without HP1α, controlled by the addition of auxin. Mitotic chromosomes isolated from cells treated with methylstat to increase methylated histone levels indeed show a significant, greater than 100% increase in doubling force from 262 ± 50 pN to 745 ± 164 pN (p=0.005, *Figure 5*, D and E), recapitulating previous results for HeLa cells (*Biggs et al., 2019*). Mitotic chromosomes isolated from cells treated with both methylstat to increase methylation and

auxin to degrade HP1α have a doubling force comparable to those treated with methylstat alone, 452 ± 116 pN (p=0.18, *Figure 5*, D and E). Oppositely, mitotic chromosomes from cells that were treated only with auxin compared to both auxin and methylstat had significantly different doubling forces (148 vs. 452 pN, p=0.005). The data suggest that histone methylation stiffens mitotic chromosomes independently of HP1α and thus has a dominant role in determining mitotic chromosome mechanics. At the same time, we emphasize that HP1α clearly plays a major role in mitotic chromosome mechanics in wild-type cells.

HP1α depletion is known to lead to chromosomal instability, aberrant recombination, anaphase bridges, and lagging chromosomes (*Chu et al., 2014*). Therefore, HP1α's role in metaphase chromosome mechanics may have functional importance during mitosis. To test this, we measured the percentage of mitotic cells with chromosome misalignment in metaphase or anaphase bridges during chromosome segregation in control, 4 hr, and 16 hr auxin-treated HP1α-degraded populations. HP1α depletion resulted in significant increases in both metaphase misalignment, from 8% in control to 28% in auxin 4 hr and 41% in auxin 16 hr treatments, and mis-segregation as measured by ana/telophase bridges, from 15% to 28% in auxin 4 hr and 50% in auxin 16 hr treatments (all p<0.05, *Figure 5F*). Thus, loss of HP1α disrupts chromosome mechanics and causes dysfunction in mitosis via chromosome misalignment and mis-segregation.

Abnormal mitosis has also been reported to disrupt nuclear morphology in the daughter cells (*Gisselsson et al., 2001*). Thus, we tracked cells treated without or with auxin for 24 hr through mitosis to determine if abnormal mitosis resulted in abnormal nuclear morphology after mitosis. Abnormal mitosis in parental or untreated HP1α-AID-sfGFP cells is rare, but it results in daughter cells with high nuclear curvatures (red dots, *Figure 5G*). Cells with HP1α degraded more frequently undergo abnormal mitosis (*Figure 5*, F and G). Interestingly, following both normal and abnormal mitosis, HP1α-degraded daughter cells exhibit increased average nuclear curvature in G1, 4 hr after mitosis (*Figure 5G*). This data suggests abnormal mitosis upon HP1α degradation may not be the primary cause of abnormal nuclear shape since normal mitosis under these conditions results in equally high curvature for daughter nuclei after mitosis (*Figure 5G*). Taken together, HP1α is necessary for proper mitotic chromosome mechanics and function, and its depletion results in abnormal mitosis and, independently, abnormally shaped daughter interphase nuclei.

## Discussion

Constitutive heterochromatin comprises an essential nuclear compartment known to perform genome-stabilizing functions through its biochemical and mechanical properties. HP1α is an essential protein component of heterochromatin that orchestrates its structural and functional roles (*Kumar and Kono, 2020*). To directly characterize these roles, we developed a new tool for rapid and reversible depletion of endogenous HP1α protein through auxin-inducible degradation (*Nishimura et al., 2009*). Interestingly, rapid degradation of HP1α over 4 hr does not significantly alter large-scale transcriptional profile or chromatin organization (*Figure 1*). Nonetheless, rapid degradation of HP1α has significant effects on interphase and mitotic chromosome mechanics and morphology (*Figures 2–4*). Furthermore, HP1α's role is dependent on its ability to dimerize (*Figure 3*). Together with polymer simulations of interphase nuclear mechanics (*Figure 4*), these results indicate that HP1α acts as a dynamic chromatin-chromatin crosslinker to provide mechanical strength to the nucleus, and that this function may persist through mitosis.

### HP1α is not essential for transcription repression or heterochromatin maintenance on short time scales

Our data are the first to separate the direct and indirect roles of HP1α in heterochromatin and its major functions in maintaining heterochromatin and regulating transcription. Early studies of HP1α established its association with compacted regions (beta chromatin) (*Bannister et al., 2001*), transcriptional silencing in yeast (*Fischer et al., 2009*; *Sadaie et al., 2008*), and silencing in *Drosophila* and mammalian cells at specific sites (*Li et al., 2003*; *Verschure et al., 2005*). Recent studies have shown a capacity for HP1α to suppress transcription in HEK293 cells when overexpressed (*Lee et al., 2019*) and in MEF cells while recruited to a specific array (*Erdel et al., 2020*). In contrast, our studies assay global transcription after rapid loss of endogenous HP1α in human cells. We find that rapid HP1α degradation does not result in significant changes in gene transcription and local

compaction (*Figure 1* and *Figure 1—figure supplement 1*), suggesting that its presence is dispensable for maintenance of these heterochromatic features over timescales < 24 hr.

Chromatin compaction and transcriptional repression also depend on methylation, which promotes HP1α binding, which in turn may recruit the methyltransferases for further propagation of methylation (*Bannister et al., 2001*). However, we found that HP1α is not necessary for short-term (4 hr) or longer-term (16 hr) maintenance of histone methylation. In particular, after rapid degradation of HP1α by our endogenous auxin-induced degradation construct, there was no significant change in constitutive heterochromatin marker H3K9me$^3$, commonly associated with transcriptional repression. The lack of widespread changes in transcription agrees with the lack of change in H3K9me$^3$ levels (*Figure 2*). This is consistent with a previous report that genetic deletion of HP1α does not alter global H3K9me$^3$, but rather, alters specific satellite H3K9me$^3$ in parts of the genome with repetitive DNA sequences (*Bosch-Presegué et al., 2017*). Furthermore, our results are supported by the recent finding that heterochromatin foci sizes, compaction, and accessibilities are independent of HP1α binding (*Erdel et al., 2020*). Together, these results indicate an inability for the transcription machinery to function at heterochromatic loci regardless of whether or not HP1α is present. Our data also showed that increased histone methylation via methylstat did not result in a global increase in HP1α levels.

Altogether, these results are consistent with the existence of a heterochromatin compaction state that is insensitive to the presence or absence of HP1α (*Erdel et al., 2020*). Instead, the functional impact of HP1α may appear in other processes, such as DNA replication (*Schwaiger et al., 2010*), chromosome segregation (*Abe et al., 2016*), epigenetic imprinting and inheritance (*Hathaway et al., 2012*; *Holla et al., 2020*; *Nakayama et al., 2000*), or post-mitotic reformation of the nucleus (*Liu and Pellman, 2020*). Nonetheless, as we discuss below, despite its limited impact on global transcription and chromatin organization, HP1α serves an important function as a mechanical stabilizer of the genome and nucleus.

## HP1α governs nuclear stiffness with a distinct and separate mechanical contribution from histone methylation

While rapid depletion of HP1α did not alter heterochromatin-specific properties and functions such as histone methylation levels or transcriptional repression over short time scales, it did significantly contribute to nuclear mechanics. Degradation of HP1α resulted in a drastic decrease in the short-extension rigidity of the nucleus, reducing the spring constant by 45% (*Figure 2B,D*). Lamin A levels and large-deformation nuclear stiffness, however, were unaffected by HP1α degradation (*Figure 2B* and *Figure 2—figure supplement 1A*). These results are consistent with prior experiments showing that chromatin dominates the mechanical response to small deformations, while lamins underlie strain stiffening to large deformations (*Stephens et al., 2017*). Similarly, HP1α has been shown to provide mechanical resistance for a single DNA fiber (*Keenen et al., 2021*). Furthermore, consistent with HP1α's newfound role in chromatin-based mechanics, we find that HP1α degradation results in the loss of nuclear shape stability (*Figure 2E,G*), similar to the effects of other chromatin perturbations that soften the cell nucleus (*Furusawa et al., 2015*; *Stephens et al., 2019a*; *Stephens et al., 2019b*; *Stephens et al., 2018*; *Wang et al., 2018*). Thus, while rapid depletion of HP1α has little apparent effect on genome organization (*Figure 1*), HP1α is critical to maintaining the mechanical integrity of chromatin.

It is known that the mechanical contribution of chromatin to the short-extension force response of the nucleus depends on histone modification state (*Heo et al., 2016*; *Hobson and Stephens, 2020b*; *Krause et al., 2019*; *Liu et al., 2018*; *Nava et al., 2020*; *Stephens et al., 2019b*; *Stephens et al., 2018*; *Stephens et al., 2017*). We considered the possibility that histone methylation contributes to mechanics through its impact on HP1α binding to chromatin (*Bannister et al., 2001*; *Erdel et al., 2020*; *Lachner et al., 2001*; *Nakayama et al., 2001*). However, our experiments show that histone methylation has a distinct contribution to chromatin-based nuclear mechanical response that is largely separate from HP1α (*Figure 2D,F,G*). In particular, nuclear rigidity (and corresponding shape stability) lost by HP1α degradation can be recovered by hypermethylation of histones via methylstat treatment. Furthermore, HP1α has an additive effect with methylation on nuclear mechanical response: chromatin-based nuclear stiffness decreases after HP1α degradation with or without treatment with methylstat. Together, these results suggest that HP1α and histone methylation modulate separable mechanical responses within the cell nucleus. The methylation-

based mechanical response may be due to direct interactions between histone marks (*Bilokapic et al., 2018*; *Zhiteneva et al., 2017*) or effects of other histone mark readers.

## HP1α contributes to nuclear mechanical response by acting as a chromatin crosslinker

What is the separate mechanical role of HP1α in heterochromatin? HP1α is a homodimer capable of physically bridging chromatin fibers by binding methylated histones or DNA (*Canzio et al., 2011*; *Cheutin et al., 2003*; *Machida et al., 2018*). We found evidence that this capability supports a distinct mechanical function. HP1α's dimerization is essential to its role in maintaining nuclear shape stability (*Figure 3*), which has been shown here (*Figure 2*) and previously (*Stephens et al., 2019a*; *Stephens et al., 2018*) to depend on chromatin-based nuclear stiffness. Thus, we conclude that HP1α's ability to dimerize and crosslink chromatin is essential to HP1α's contributions to chromatin-based nuclear stiffness (*Figure 6*).

This interpretation is supported by coarse-grained polymer simulations of cell nuclear mechanical response. In our model, chromatin is modeled as a crosslinked polymer gel, while the nuclear lamina is modeled as a polymeric shell that is physically linked to the interior chromatin. This model previously recapitulated measurements from nucleus micromanipulation experiments, which observed the two-regime force-extension relationship, its dependence on histone modifications (altering chromatin fiber stiffness) and nuclear lamins (altering the polymeric shell meshwork), and the changes to the shape of the nucleus when it is stretched (*Banigan et al., 2017*; *Stephens et al., 2017*). Here, we showed that the short-extension stiffness, but not the long-extension stiffness, is highly sensitive to the number of chromatin-chromatin crosslinks (*Figure 4C,D*). Specifically, with few crosslinks, the

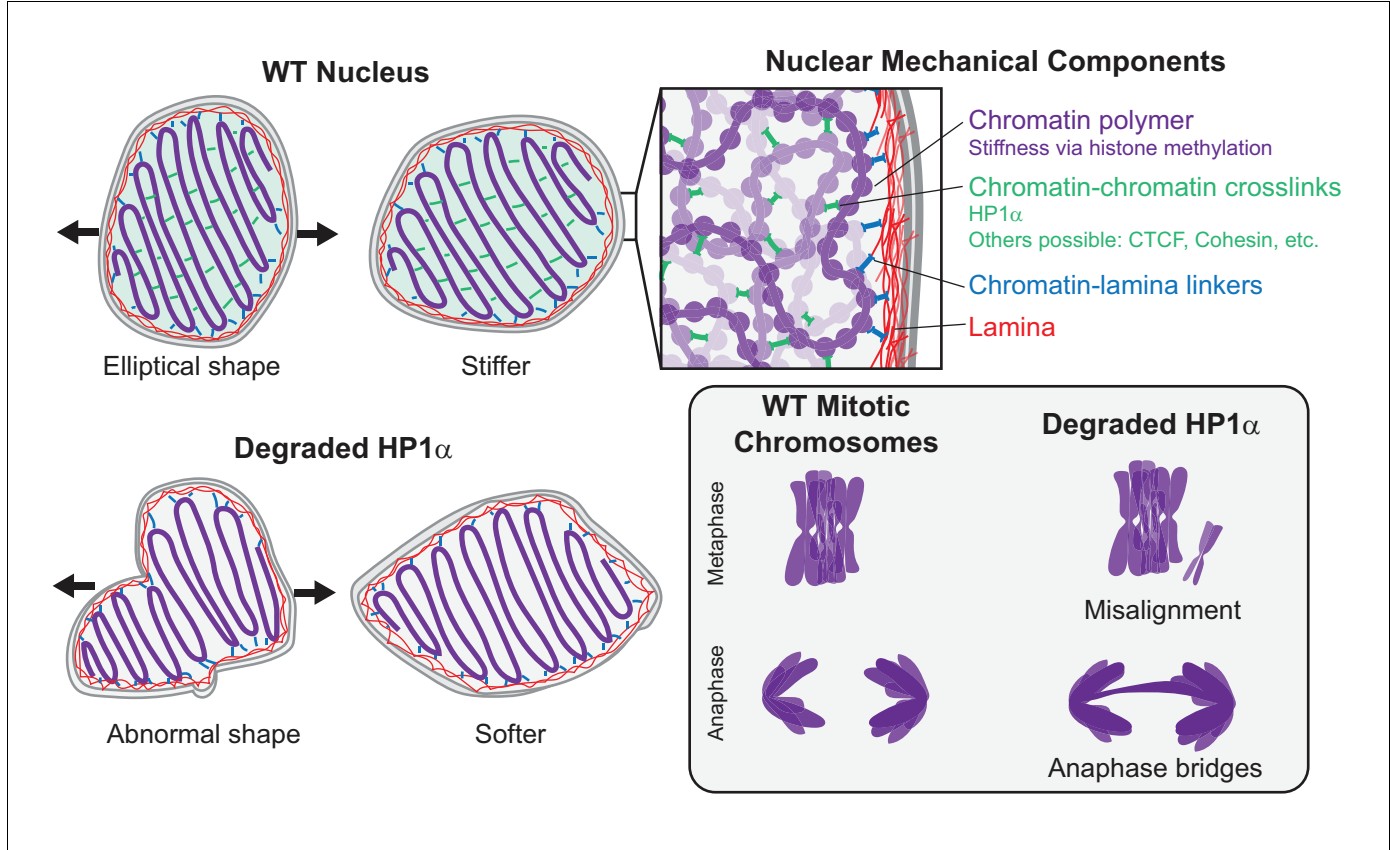

**Figure 6.** HP1α is a mechanical element of interphase nuclei and mitotic chromosomes. In wild-type (WT) nuclei, HP1α acts as a chromatin-chromatin crosslinker, resulting in stiffer nuclear mechanics. Other components that contribute to nuclear mechanics include the chromatin polymer (whose mechanical contribution is dictated by histone methylation), the lamina, and chromatin-lamina linkages. Nuclei with HP1α degraded have abnormal shape and softer chromatin-based short-extension mechanical response. Degradation of HP1α also leads to softer mitotic chromosomes and mitotic defects, including chromosome misalignment and anaphase bridges.

short-extension spring constant is small (~50% of the WT simulation), which parallels the observed result for nuclei with HP1α degraded (*Figure 2*) and the expected mechanics for nuclei with the HP1α dimerization mutant (see above). Thus, the separate roles of histone methylation and HP1α can be modeled as altering the chromatin polymer fiber and chromatin-chromatin crosslinks, respectively (*Figure 6*). Interestingly, although we model HP1α as a permanent chromatin-chromatin crosslink, chromatin binding by HP1α in vivo is transient, with a typical exchange time of ~10 s (*Cheutin et al., 2003*; *Festenstein et al., 2003*; *Kilic et al., 2015*). Apparently, chromatin-bound HP1α is sufficiently abundant that crosslinks continuously percolate interphase chromatin to provide a robust mechanical response and thereby maintain nuclear shape. Simulations with transient crosslinks or experiments with HP1α chromatin-binding mutants (*Nielsen et al., 2001*) could further investigate this phenomenon and its implications for chromatin organization, chromatin-based nuclear mechanics, and nuclear morphology.

More broadly, the finding that HP1α acts as a chromatin crosslinker is consistent with other experimental data suggesting that chromatin organization and mechanics is supported by widespread physical crosslinking. Recent chromosome conformation capture (Hi-C) and micromanipulation experiments show that moderate fragmentation of chromatin does not alter genome organization and mechanics. On the basis of these experiments, it is hypothesized that chromatin may be physically linked as frequently as once per 10–25 kb (*Belaghzal et al., 2021*). Our data show that HP1α is one of the likely many possible chromatin crosslinking elements in the genome. There is a growing list of chromatin proteins and nuclear components contributing to maintenance of nuclear morphology, some of which have been identified by a genetic screen for effects on nuclear morphology (*Tamashunas et al., 2020*) and a variety of other experiments (reviewed in *Stephens et al., 2019a*). Other chromatin crosslinkers to be investigated include chromatin looping proteins and other components implicated by various experiments, such as cohesin, CTCF, mediator, and possibly RNA.

Crosslinking and gelation are intimately coupled to phase separation (*Harmon et al., 2017*). Therefore, HP1α may contribute to nuclear mechanics through a phase transition mechanism. In a phase transition model, HP1α dimers crosslinking certain regions of the chromatin polymer would lead to polymer-polymer or sol-gel transitions (*Khanna et al., 2019*; *Tanaka, 2002*) that contribute to the elastic modulus of the whole network (*Colby and Rubinstein, 2003*; *Semenov and Rubinstein, 2002*; *Shivers et al., 2020*). Furthermore, HP1α binding to methylated histones is known to alter the structure of the nucleosome core, which could promote nucleosome-nucleosome interactions, and induce polymer-polymer phase separation of the chromatin fiber (*Sanulli et al., 2019*). Additionally, purified HP1α protein in vitro exhibits liquid-liquid phase separation by itself, with naked DNA, and with nucleosome arrays (*Larson et al., 2017*; *Shakya et al., 2020*), and HP1 condensates bound to dsDNA in vitro can lend mechanical strength (*Keenen et al., 2021*). The material properties of these in vitro condensates varies depending on the chromatin content (*Larson et al., 2017*; *Shakya et al., 2020*). More generally, phase separation in an elastic network such as chromatin can be regulated by the local mechanical properties of the material (*Shin et al., 2018*; *Style et al., 2018*). Together, these observations suggest a complex physical picture that is dictated by both HP1α's self-interaction and chromatin binding capabilities, in addition to length, concentration, and phase behavior of the chromatin itself (*Gibson et al., 2019*; *Maeshima et al., 2021*; *Strickfaden et al., 2020*). The material state and categorization of the phase transition of HP1α-rich heterochromatin in vivo have been debated (*Erdel et al., 2020*; *Larson et al., 2017*; *Strom et al., 2017*; *Williams et al., 2020*), and the underlying chromatin may be 'solid-like' (*Strickfaden et al., 2020*), so further work is necessary to completely understand the interplay of these components in determining phase behavior and mechanics of the interphase nucleus.

## HP1α is a mechanical element of mitotic chromosomes and is essential for proper mitosis

Mechanical components of interphase chromatin may remain attached to mitotic chromosomes in order to maintain the mechanical strength of chromosomes during mitosis. Recent work has shown that heterochromatin-based histone modifications/methylation also control the mechanical strength of chromosomes, while euchromatin-based histone acetylation does not (*Biggs et al., 2019*). That paper hypothesized that increased histone methylation could be aided by 'histone reader' heterochromatin-associated proteins, specifically HP1α. Our data reveal that, similar to HP1α in interphase

nuclei, HP1α during mitosis is a significant mechanical component of the mitotic chromosome (*Figure 5*). HP1α degradation leads to more extensible mitotic chromosomes, but the stiffness can be recovered by hypermethylation via methylstat treatment. The fact that HP1α still provides mechanical stiffness in mitotic chromosomes, a chromatin-only system without lamins, further supports that HP1α mechanically functions as a chromatin crosslinker. Previous work has proposed that mitotic chromosomes are dense polymer gels based on their elastic response, which relies on the continuity of the DNA backbone (*Poirier and Marko, 2002*), topology (*Kawamura et al., 2010*), and the chromatin cross-bridging condensin protein complex (*Sun et al., 2018*). Our experiments implicating HP1α as a crosslinking element (in interphase) and measuring the mechanical contributions of HP1α in mitotic chromosomes further support this picture. Methylation could serve as an additional compaction agent by providing further crosslinking, stiffening the chromatin fiber itself, or generating poor solvent conditions that further compact mitotic chromosomes (*Batty and Gerlich, 2019*; *Gibcus et al., 2018*; *Maeshima et al., 2018*). Together, these components generate the rigidity necessary for robust mitotic chromosomes.

Loss of HP1α results in dysfunction, marked by improper chromosome alignment and segregation. Previous reports had noted that loss of HP1α and HP1γ, specifically at the centromere, causes increased incidence of chromatin bridges (*Lee et al., 2013*) and mitotic alignment errors (*Yi et al., 2018*), genetic deletion of HP1α increases merotelic and syntelic attachments (*Bosch-Presegué et al., 2017*), and mitosis is dependent on HP1α phosphorylation (*Chakraborty et al., 2014*). Our findings with rapid degradation of HP1α reveal a threefold increase in both misalignment and missegregation, which were mostly observed as anaphase bridges, which could be due, in part, to aberrant DNA damage repair (*Chiolo et al., 2011*; *Peng and Karpen, 2007*). Our results are in agreement with HP1α interacting with LRIF1 at the centromere, which when perturbed results in similar misalignment and missegregation (*Akram et al., 2018*). However, further work is required to determine if chromosome misalignment is due to a biochemical pathway or mechanical pathway where whole-chromosome mechanics controlled by HP1α influences proper segregation.

## Conclusion

We have established that HP1α has consistent mechanical and functional implications for chromosomes throughout the cell cycle. While rapid degradation of HP1α has little effect on the global transcriptional profile, loss of HP1α strongly impairs interphase and mitotic chromosome mechanics. This leads to deleterious and potentially catastrophic effects, such as abnormal nuclear morphology and chromosome segregation defects. When present, HP1α is a crosslinking element, and it mechanically stabilizes interphase and mitotic chromosomes, suppressing abnormal nuclear deformations and mitotic defects. It remains unclear whether HP1α's phase separation capability is important to this biophysical function. More broadly, our experiments demonstrate that mechanical softening of the nucleus due to loss of HP1α's chromatin crosslinking ability, rather than transcriptional changes, could underlie defects in fundamental nuclear functions such as nuclear compartmentalization, DNA damage prevention and response, and migration, all of which have been shown to depend on nuclear mechanics (*Gerlitz, 2020*; *Stephens, 2020*; *Xie et al., 2020*). These mechanical changes could also have broad implications for human diseases, such as breast cancer, where increased invasiveness (migration ability) has been correlated with decreased HP1α levels (*Vad-Nielsen and Nielsen, 2015*) and inhibition of HP1α dimerization (*Norwood et al., 2006*). Overall, we have revealed a direct structural role for HP1α in whole-nucleus and mitotic chromosome mechanics that furthers our understanding of chromatin-based nuclear stiffness and has important cellular functional consequences.

## Materials and methods

**Key resources table**

| Reagent type (species) or resource | Designation | Source or reference | Identifiers | Additional information |
|---|---|---|---|---|

*Continued on next page*

*Continued*

| Reagent type (species) or resource | Designation | Source or reference | Identifiers | Additional information |
|---|---|---|---|---|
| Gene (*H. sapiens*) | CBX5 | GenBank | | |
| Cell line (*H. sapiens*) | U20S | ATCC | ATCC HTB-96 | RRID:CVCL_0042 |
| Transfected construct (*H. sapiens*) | 3' HP1α-AID-sfGFP 2A PuroR | Addgene | 127906 | RRID:Addgene_127906 |
| Transfected construct (*H. sapiens*) | Guide RNA/Cas9 plasmid pX330 human 3' HP1α gRNA | Addgene | 127906 | RRID:Addgene_127906 |
| Sequence-based reagent | Guide RNA sequence, 5'-acagcaaagagctaaaggag −3' | This paper | | |
| Transfected construct (*H. sapiens*) | HP1α-mCherry | This paper, in pHR vector | | HP1α PCR from Addgene_17652 |
| Transfected construct (*H. sapiens*) | HP1αI165E- mCherry | This paper, in pHR vector | | Point mutant made with quickchange |
| Antibody | Anti-HP1 alpha (rabbit monoclonal) | Abcam | ab109028 | (1:250) RRID:AB_10858495 |
| Antibody | anti-H3K9me2/3 (mouse monoclonal) | Cell Signaling | 5327 | (1:100) RRID:AB_10695295 |
| Antibody | anti-Lamin B1 (rabbit polyclonal) | Abcam | ab16048 | (1:500) RRID:AB_443298 |
| Antibody | anti-Lamin A/C (mouse monoclonal) | Active Motif | 39287 | (1:1000) RRID:AB_2793218 |
| Chemical compound, drug | Auxin (NaIAA) | Sigma | I5148 | 1 mM |
| Chemical compound, drug | Methylstat | Sigma | SML0343-5MG | 1 µM |
| Software, algorithm | Kappa, nuclear curvature | FIJI | *Schindelin et al., 2012* | |

## Cell lines, cloning, and characterization of HP1α-AID-sfGFP degron clone

U2OS (ATCC HTB-96) were validated by STR. These cells were cultured in DMEM/FBS and co-transfected with two plasmids, human 3' HP1α-AID- sfGFP 2A PuroR (Addgene 127906) and a guide RNA/Cas9 plasmid pX330 human 3' HP1α gRNA (Addgene 127907) with Lipofectamine 2000 according to manufacturer's instructions. The guide RNA sequence, 5'- acagcaaagagctaaaggag −3', flanked the stop site of the CBX5 gene and was destroyed upon successful in-frame insertion of the AID-GFP 2A PuroR cassette. Modified cells were selected with 10 µg/ml puromycin and single-cell sorted into 96-well plates with a BD FACS Aria III gated with FACSDiva software to sort only the top 10% brightest GFP-expressing cells. Expression of HP1α-AID-sfGFP was monitored by fluorescence microscopy as clones were expanded and subjected to quality control (QC; quality control, consisting of immunoblotting, PCR and live cell microscopy, see supplementary materials). A homozygous clone that passed all QC (U2OS HP1α 4) was co-transfected with the transposon vector pEF1a-OsTIR-IRES-NEO-pA-T2BH (Addgene 127910) and SB100X in pCAG globin pA (Addgene 127909). Forty-eight hr post-transfection, cells were selected with 400 µg/ml G418 for 10 days (media with fresh G418 replaced every 2–3 days) and then allowed to recover in DMEM/FBS for 1 week. GFP positive cells were again single cell sorted, expanded and subjected to QC. Degradation of HP1α-AID-sfGFP by OsTIR1 was evaluated by flow cytometry, immunoblotting and live cell microscopy after treatment with 1 mM auxin (NaIAA, Sigma #I5148) for 4–16 hr. A clone (U2OS HP1α 4–61) that

by all QC measures demonstrated no detectable HP1α-AID-sfGFP after auxin treatment was chosen and expanded.

## Validation by PCR

Genomic DNA was extracted using the PureLink Genomic DNA Mini Kit (catalog number K182001) and PCR was performed with oligos that flanked the insertion site, yielding 2 PCR products for heterozygous HP1α clones or a single larger PCR product for HP1α clones homozygous for the AID-GFP-Puro insertion.

## Cell line validation microscopy

Live cells were plated into four chambered glass bottomed dishes (Greiner Bio One, #627975) and mounted in a temperature and $CO_2$ controlled chamber (Okolab) for viewing using a Nikon Eclipse Ti inverted microscope with a 100X, 1.45NA phase objective and Spectra X (Lumencor) LED excitation at DAPI (395/25) and GFP (470/24) wavelengths (used at 5% power). Cells grown on glass coverslips, fixed in 4% paraformaldehyde (Polysciences, #18814) and mounted in Prolong Diamond to preserve GFP signal were also prepared. Images were captured using an Orca Flash 4 sCMOS camera and analyzed, cropped and contrast adjusted for display using either Elements or Imaris software. Cells were tested for mycoplasma via imaging using hoechst weekly.

## Immunoblotting and immunostaining

Cell pellets from each clone were resuspended and incubated in RIPA buffer (Thermo Scientific # 89901) containing 2x Protease inhibitor (Thermo Scientific # A32955) for 1 hr on ice, and then incubated for 10 min at RT with 25U benzonase nuclease (Millipore Sigma 70746-10KUN)/50 μL sample. After BCA protein quantification (Pierce), samples were subjected to reducing SDS-PAGE and LI-COR Western blot analysis. Anti-HP1 alpha primary antibody (Abcam #ab109028) was used at 1:250 and IRDye 680CW secondary (LiCOR #925–6807) was diluted 1:15000. Blots were scanned on an Odyssey CL$_x$. Immunostaining was carried out as previously described (*Politz et al., 2002*) using Abcam #ab109028 primary antibody at 1:250, and secondary antibody (Jackson labs 711-165-152) at 1:200, and coverslips were mounted in Prolong Gold. Images were captured as described above.

## RNA-seq

RNA was isolated using the Qiagen RNeasy kit according to manufacturer's instructions. Cells were homogenized with a QIAshredder (Qiagen #79654) with β-mercaptoethanol in the RTL buffer. DNA was digested with RNase-Free DNase (Qiagen #79254) and purified with RNeasy MinElute Cleanup Kit (Qiagen #74204). Purified RNA was quantitated with a Nanodrop spectrophotometer and quality was confirmed on a bioanalyzer with a TapeStation R6K assay. A sequencing library from RNA with a RIN >9.5 was prepared using the TruSeq stranded mRNA Library Prep and sequencing was performed using an Illumina HiSeq 2500 workstation. There were over 16,000 genes with one transcript per million reads for control compared to auxin 4 hr as well as control compared to auxin 16 hr, The RNA-Seq reads were mapped with STAR and then quantified by RSEM, and the differential gene expression analysis was performed using DESeq2.

## HP1α rescue constructs

Full length HP1α was amplified by PCR (Addgene 17652), and cloned using InFusion kit into a pHR lentiviral vector under an SFFV promoter and tagged C-terminally with mCherry and sspB. A mutation was introduced to disrupt dimerization at amino acid 165 in the chromoshadow domain, changing the codon ATA (coding for Isoleucine, I) to GAG (coding for Glutamic acid, E) to result in HP1α$^{I165E}$. This mutation has been previously characterized to disrupt homodimerization of HP1α (*Brasher et al., 2000*).

## Lentiviral expression of HP1α rescue constructs

LentiX cells were transfected with transfer plasmids pCMV-dR8.91 and pMD2.G, as well as expression construct of interest in a 9:8:1 mass ratio into HEK293T cells using FuGENE HD Transfection Reagent (Promega) per manufacturer's protocol. After 48 hr, media containing viral particles was collected and filtered using 0.45 micron filter (Pall Life Sciences), and either used immediately or stored

at −80°C. HP1α-AID-sfGFP cells were plated at 15–20% confluency on glass-bottom 96-well plates (Cellvis) and infected with 10–50 μL of virus-containing media. After 24 hr, viral media was removed and replaced with fresh DMEM, and cells were fixed or imaged at 3–7 days post-infection.

## Immunostain, microscopy, and morphological analysis of nuclear shape in fixed cells

HP1α-AID-sfGFP cells expressing mCherry-tagged HP1α$^{WT}$ or HP1α$^{I165E}$ were treated with control media (no auxin) or 1 mM auxin (NaIAA, Sigma #I5148) for 16 hr before being fixed in 4% paraformaldehyde for 10 min, washed three times in PBS, permeabilized with 1% triton X-100 in PBS for 1 hr at room temperature with rocking, blocked with 5% FBS in 0.25% PBST for 1 hr at room temperature with rocking, and incubated with anti-Lamin A/C antibody 1:1000 (Active Motif, 39287) in block overnight. Samples were washed again three times with PBS and incubated with Goat anti-mouse IgG secondary antibody conjugated to Alexa fluor 647 (Thermo Fisher, A-21236) for >2 hr, washed again and incubated with Hoechst 1:2000 in PBS for 30 min. Images of fixed and stained cells were obtained with a spinning-disk confocal microscope (Yokogawa CSU-X1) with 100X oil immersion Apo TIRF objective (NA 1.49) and Andor DU-897 EMCCD camera on a Nikon Eclipse Ti body. Live samples were maintained at 37°C and 5% $CO_2$ by a 96-well plate incubation chamber (Okolab). 405, 488, 561, and 647 lasers were used for imaging Hoechst, sfGFP, mCherry or Alexa 568, and Alexa 647, respectively. Laser power and digital gain were consistent for imaging all samples across an experiment, allowing for quantitative comparison of fluorescent intensities. Morphological analysis was performed in FIJI using a plugin that measures curvature; Kappa (*Schindelin et al., 2012*), which was created originally by Kevan Lu and is now maintained by Hadrien Mary. Briefly, one z-slice of the Lamin A/C immunostain channel at the center of the height of the nucleus was loaded into the Kappa plugin, traced, and a closed curve was fit to the signal. Curvature along the nuclear envelope trace was calculated as the inverse radius of curvature with the plugin and an average value of curvature per nucleus was recorded.

## Peripheral association of chromatin

HP1α-AID-sfGFP cells were treated with control media or 1 mM Auxin for 16 hr, then immunostained with α-H3K9me2/3 (Cell Signaling, mouse mAb #5327) and α-Lamin B1 (Abcam ab16048), as described above, and fixed-cell images obtained as above. Lamin B1 stain was used in FIJI to define a nuclear periphery mask, and enrichment was calculated as average intensity within the periphery mask divided by average intensity of the nuclear interior.

## Microscopy and morphological analysis of nuclear shape in living cells

Control media (no auxin) or 1 mM auxin (NaIAA, Sigma #I5148) was added to HP1α-AID-sfGFP cells expressing an miRFP-tagged histone H2B plated in 96-well glass bottom plates (Cellvis). Twenty-five X-Y points were chosen in each of the control and experimental wells, and a z-stack ranging eight microns was collected at each point every 30 min for 12 hr (with auxin added to experimental wells at time 0 hr). Morphological analysis was again performed with the FIJI plugin Kappa, this time using the histone signal to delineate the edge of the nucleus.

## Nuclear rupture and DNA damage analysis

HP1α-AID-sfGFP U2OS cells were grown in cell culture dishes containing glass coverslip bottoms (In Vitro Scientific). Cells were treated 1 or 2 days prior to imaging with Cell Light Nucleus-RFP (NLS-RFP Fisher Scientific). Cells were untreated or treated with auxin 4–6 hr prior to imaging. Cells were imaged with a 40 × 0.75 NA air objective on an environmental incubation (37°C and 5% CO2) at 2 min intervals for 3 hr. Nuclear ruptures by observing RFP spilling out of the nucleus and into the cytoplasm as outlined in *Robijns et al., 2016*. DNA damage foci were counted using the Immunostaining procedure above with γH2AX conjugated Alexa 657 antibody (CST 9720, 1: 300) and Elements Bright Spot detection to determine the number of foci.

## Cell protocol for single nucleus and mitotic chromosome isolation

Micromanipulation experiments used U2OS parent or HP1α-AID-sfGFP cells maintained in DMEM (Corning) with 10% fetal bovine serum (FBS) (HyClone) and 1% 100x penicillin streptomycin

(Corning). The cells were plated and allowed to recover 1–3 days before nucleus or chromosome isolation. 1 mM auxin (NaIAA, Sigma #I5148) was added 4–6 hr before nucleus and chromosome isolation in '+auxin' and '+auxin+methylstat' experiments, and 1 µM methylstat was added 30–38 hr before '+methylstat' and '+auxin+methylstat' experiments. All experiments were performed without synchronization.

## Single nucleus and mitotic chromosome isolation

Single nucleus (*Stephens et al., 2017*) and chromosome isolation (*Biggs et al., 2019*) experiments were performed on an inverted microscope (IX-70; Olympus) with a 60 × 1.42 NA oil immersion objective with a ×1.5 magnification pullout. Nuclei and Chromosomes were isolated at room temperature and atmospheric $CO_2$ levels in DMEM 10% FBS 1% pen strep media in 3 hr or less to ensure minimal damage to the cells and chromosomes. Before isolation all cells were imaged for the absence or presence of HP1α with or without auxin treatment, respectively. For nucleus isolation, cells were treated with 1 µg/mL latrunculin A (Enzo Life Sciences) for 45 min before isolation to depolymerize the actin cytoskeleton. Interphase cells were lysed with 0.05% Triton-X 100 in PBS. After lysis, micromanipulation pipettes filled with PBS were used to capture and position the single isolated nucleus. Isolation aimed for G1 nuclei determined by their size (10–15 µm along the major axis). For chromosomes, prometaphase mitotic cells were identified by eye and lysed with 0.05% Triton-X 100 in PBS. After lysis, the bundle of interconnected chromosomes fell out of the cell and stabilized with a PBS filled pipette by light aspiration. While the bundle was stabilized, one end of a loose chromosome was aspirated into an easily bendable (Kavg = 40 pN/µm) 'force' pipette, moved away from the bundle, where the other end of the chromosome was grabbed by a stiff pipette. The bundle was heavily aspirated into the stabilizing pipette and then removed, leaving an isolated chromosome to be manipulated.

## Nucleus mechanics measurements

The isolated nucleus is suspended between a stiff pull pipette and a pre-calibrated force pipette for defined size (2.8–3.3 µm diameter) and bending constant (1.5–2.0 nN/µm). The pull pipette provides either 3 µm extension (short regime only) or 6 µm extension of the nucleus (long regime) at a rate of 0.05 µm/s. Bending of the force pipette relative to extension of the nucleus provides a measure of force. Data is transferred to Excel where the slope of the force-extension provides a nuclear spring constant for chromatin (short extension 0–3 µm extension) and a lamin-based strain-stiffening nuclear spring constant (long regime slope minus short regime slope).

## Mitotic chromosome mechanics measurements

Once a mitotic chromosome was isolated, the stiff pipette was moved 6.0 µm at a rate of 0.20 µm/s with step sizes of 0.04 µm/step using a Labview program, while the force pipette (Fp) and stiff pipette (Sp) were visually tracked. A linear regression of the deflection vs stretch (Fp/(Sp-Fp)) slope was calculated, multiplied by the force pipette spring constant (calibrated after the experiment) to give the spring constant of the chromosome, and multiplied by the initial length of the chromosome, to give the doubling force of the chromosome in a custom Python script, which is publicly available on GitHub (https://github.com/ebanigan/doubling_force) (*Shams and Biggs, 2021*).

## Mitotic chromosome fluorescence

Cells were imaged on an inverted microscope (IX-70; Olympus) with a 60 × 1.42 NA oil immersion objective with a ×1.5 magnification pullout. in the GFP channel once a mitotic cell was identified and the final isolated chromosome was imaged in the GFP channel for each experiment to determine if they contained HP1α. Periodically, the chromosome bundle was also imaged in the GFP channel.

## Nuclear morphology solidity measurements

Nuclei were selected via intensity threshold in Hoechst channel and made into an object or ROI and reported for shape solidity, which is a ratio of area over convex area of the nucleus. The threshold of 0.96 solidity was used to determine normal versus abnormally shaped nuclei.

## Brownian dynamics simulations

Brownian dynamics simulations of a polymeric shell linked to an interior crosslinked polymer were performed as described previously (*Banigan et al., 2017*; *Stephens et al., 2017*). Simulation source code is publicly available on GitHub (https://github.com/ebanigan/shell-polymer) (*Banigan, 2021*). A total of 1000 shell subunits with diameter $a_s = 0.71$ µm are randomly placed on a sphere of radius $R_i = 10$ µm, which is shrunk to $R_0 = 5$ µm during the simulation initialization (*Banigan et al., 2017*). Each shell subunit is connected by springs to $4 \leq z \leq 8$ nearest neighbor shell subunits ($<z> \approx 4.5$). A linear polymeric chain of 552 subunits with diameter $a_p = 0.57$ µm, connected by springs, is initialized in a random globular conformation within the shell. The polymer is randomly crosslinked with $N_C$ crosslinks, where $N_C = 55$ (20% of all polymer subunits are crosslinked) unless noted. $N_L$ polymer subunits near the surface of the sphere are linked by springs to the nearest shell subunit; $N_L = 40$ (i.e. 7.2% of all polymer subunits or 22% of all peripheral subunits, defined by contact with the shell subunits in the initial configuration, are linked to the shell) unless noted. Tensile force is exerted across the nucleus by exerting force $F$ along the x-axis on a single shell subunit at each of the two poles.

Spring potentials governing interactions between subunits have the form $U_{sp}=(k_{sp}/2)(r_{ij}-r_{ij,0})^2$ for $r_{ij}>r_{ij,0}$, where $r_{ij}$ is the distance between subunits $i$ and $j$, $r_{ij,0}$ is the sum of the two subunit radii, and $k_{sp}$ is the spring constant, which depends on the type of potential. $k_s = 0.8$ nN/µm for shell-shell springs, $k_p = 1.6$ nN/µm for 'polymer springs' connecting subunits along the polymer backbone, $k_C = k_p$ for 'crosslink springs' connecting polymer subunits, and $k_L = k_p$ for springs linking the polymer to the shell. All subunits repel each other via soft-core excluded volume interactions, modeled as $U_{ex} = (k_{ex,ij}/2)(r_{ij}-r_{ij,0})^2$ for $r_{ij}<r_{ij,0}$, where $k_{ij}$ is the repulsive spring constant; $k_{ex,ij} = k_s$ if $i$ and $j$ are both shell subunits, $k_{ex,ij} = k_p$ if $i$ and $j$ are both polymer subunits, and $k_{ex,ij} = 2k_sk_p/(k_s +k_p)$ if one is a shell subunit and the other is a polymer subunit.

All subunits are subject to uncorrelated thermal noise (T = 300 K). The system obeys the overdamped Langevin equation, which is solved by an Euler algorithm (*Allen and Tildesley, 1987*) with timestep $dt = 0.0005$.

## Acknowledgements

This work was supported by the NIH Center for 3D Structure and Physics of the Genome of the 4DN Consortium (U54DK107980 and 1UM1HG011536) and the NIH Physical Sciences-Oncology Center (U54CA193419). CPB and ARS were supported by the Howard Hughes Medical Institute, and grants from the NIH 4D Nucleome Program (U01 DA040601); ARS is supported by the LSRF Fellowship from Mark Foundation For Cancer Research. We thank Daniel S.W. Lee for experimental discussion and support, and Yiche Chang for the generous gift of pHR-HP1α-mCherry plasmids. We thank Daniel Shams for helping write a custom script for analyzing mitotic chromosome micromanipulation force measurements. EJB was supported by the NIH Center for 3D Structure and Physics of the Genome of the 4DN Consortium (U54DK107980), the NIH Physical Sciences-Oncology Center (U54CA193419), and NIH grant GM114190. XW and FY are supported by 1R35GM124820, R01HG009906, U01CA200060 and R24DK106766. JP, DS, LT, AT, and MG were funded by 4DN (U01DA040583). KC and ADS are supported by the Pathway to Independence Award (R00GM123195) and 4D Nucleome two center grant (1UM1HG011536).

## Additional information

### Funding

| Funder | Grant reference number | Author |
|---|---|---|
| Mark Foundation For Cancer Research | Life science research foundation Postdoctoral Fellowship | Amy R Strom |
| Mark Foundation For Cancer Research | AWD1006303 | Amy R Strom |
| National Institutes of Health | U01 DA040601 | Clifford P Brangwynne |

| National Institutes of Health | GM114190 | Edward J Banigan |
| --- | --- | --- |
| National Institutes of Health | U54DK107980 | John F Marko |
| National Institutes of Health | U54CA193419 | John F Marko |
| National Institutes of Health | R24DK106766 | Feng Yue |
| National Institutes of Health | 1R35GM124820 | Feng Yue |
| National Institutes of Health | R01HG009906 | Feng Yue |
| National Institutes of Health | U01CA200060 | Feng Yue |
| National Institutes of Health | U01DA040583 | Mark Groudine |
| National Institutes of Health | 1UM1HG011536 | John F Marko<br>Andrew D Stephens |
| National Institutes of Health | R00GM123195 | Andrew D Stephens |

The funders had no role in study design, data collection and interpretation, or the decision to submit the work for publication.

## Author contributions

Amy R Strom, Conceptualization, Formal analysis, Validation, Investigation, Visualization, Methodology, Writing - original draft, Writing - review and editing; Ronald J Biggs, Conceptualization, Formal analysis, Investigation, Visualization, Methodology, Writing - review and editing; Edward J Banigan, Conceptualization, Formal analysis, Investigation, Methodology, Writing - original draft, Writing - review and editing; Xiaotao Wang, Data curation, Formal analysis, Investigation; Katherine Chiu, Formal analysis, Investigation, Visualization; Cameron Herman, Jimena Collado, Formal analysis, Validation, Visualization; Feng Yue, Mark Groudine, Supervision, Funding acquisition; Joan C Ritland Politz, Conceptualization, Resources, Data curation, Formal analysis, Validation, Methodology, Writing - review and editing; Leah J Tait, Agnes Telling, Resources, Validation, Investigation; David Scalzo, Resources, Data curation, Validation; Clifford P Brangwynne, John F Marko, Supervision, Funding acquisition, Writing - review and editing; Andrew D Stephens, Conceptualization, Data curation, Formal analysis, Supervision, Funding acquisition, Validation, Investigation, Visualization, Methodology, Writing - original draft, Project administration, Writing - review and editing

## Author ORCIDs

Amy R Strom  https://orcid.org/0000-0002-1674-3242
Ronald J Biggs  https://orcid.org/0000-0002-9965-6346
Edward J Banigan  https://orcid.org/0000-0001-5478-7425
Xiaotao Wang  http://orcid.org/0000-0002-3531-2157
Joan C Ritland Politz  https://orcid.org/0000-0001-5229-0087
Clifford P Brangwynne  http://orcid.org/0000-0002-1350-9960
John F Marko  https://orcid.org/0000-0003-4151-9530
Andrew D Stephens  https://orcid.org/0000-0001-5474-7845

## Decision letter and Author response

Decision letter https://doi.org/10.7554/eLife.63972.sa1
Author response https://doi.org/10.7554/eLife.63972.sa2

# Additional files

## Supplementary files

• Source data 1. RNAseq data analysis for HP1a-AID-sfGFP without the addition of auxin compared to auxin treatment for 4 hr. The sheet provides genes analyzed and reports log 2-Fold Change (log2-FoldChange) and q-value (padj) used to determine significance.

• Source data 2. RNAseq data analysis for HP1a-AID-sfGFP without the addition of auxin compared to auxin treatment for 16 hr. The sheet provides genes analyzed and reports log 2-Fold Change (log2FoldChange) and q-value (padj) used to determine significance.

• Source data 3. RNAseq data analysis for U2OS parental cell line compared to modified U2OS HP1a-AID-sfGFP cell line. The sheet provides genes analyzed and reports log 2-Fold Change (log2-FoldChange) and q-value (padj) used to determine significance.

• Transparent reporting form

### Data availability
We have provided the RNAseq data sets in the supplemental material as excel files.

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
