## [Decision Letter]

**Acceptance summary:**

This manuscript describes a significant and exciting role for the heterochromatin protein HP1-α in chromatin mechanics and nuclear shape stabilization that is suggested to rely on HP1-α's ability to cross-link different regions of chromatin. The work will be of much interest to communities studying chromatin biology and nuclear mechanics.

**Decision letter after peer review:**

Thank you for submitting your article "HP1α is a chromatin crosslinker that controls nuclear and mitotic chromosome mechanics" for consideration by *eLife*. Your article has been reviewed by 3 peer reviewers, and the evaluation has been overseen by a Reviewing Editor and Kevin Struhl as the Senior Editor. The following individual involved in review of your submission has agreed to reveal their identity: Sy Redding (Reviewer #2).

As you will see all reviewers found the work to be exciting and significant. Each of them also had some major concerns that need to be addressed before we can consider publication in *eLife*. Through discussion amongst the reviewers, we have outlined the essential revision recommendations below. When you submit your revision, please describe your responses to the essential revision recommendations in the context of your point-by-point responses to the individual comments by each reviewer.

Summary:

This manuscript describes a significant and exciting role for the heterochromatin protein HP1-α in chromatin mechanics and nuclear shape stabilization that may rely on the ability of the protein to cross-link different regions of chromatin. While the experiments clearly highlight the potential for such a role of HP1-α, additional work is needed to strengthen the statistical significance of the results, test the functional impact of the engineered HP1-α protein and test the role of HP1-α in chromosome mechanics during mitosis.

Essential Revisions:

1. The section of the manuscript focused on effects on mitosis is weak and should be removed or substantially improved as outlined in detail in the reviews.

2. Cell shape analyses should show actual measurement data (not phenotypic classification) and the dependency of the nuclear shape defect on mitosis should be addressed.

3. The role of HP1-lamin interaction in nuclear mechanics should be addressed.

4. Rev#2 makes the important point of validating the function of the engineered protein, which relates to concerns of Reviewer#1 on the quality (abundance) of the data showing surprising absence of effects on heterochromatin and transcription. This should be experimentally addressed or extensively discussed, as Rev#2 suggests.

5. There is concern about the small sample sizes and the differences in sample size between experimental groups of individual experiments. This concern should be addressed by increasing the sample size and making the sample sets being compared similar in size.

*Reviewer #1:*

Strom and co-authors describe a new, very intriguing role for HP1alpha in chromatin mechanics and nuclear shape maintenance which is independent of known nuclear mechanoregulators: histone methylation or nuclear lamina, and which is instead driven by HP1alpha-mediated crosslinking of chromatin. While conceptually exciting, the central conclusions lack definitive experimental evidence. Further, key measurements lack true statistical power and the polymer model predictions are not stringently challenged. Finally, the functional relevance of this HP1-mediated mechanical regulation remains mostly unexplored, as the chromatin mis-segregation phenotype is only superficially analyzed and its connection with chromatin mechanics is elusive.

1. Exclusion of contribution of chromatin methylation and lack of changes in chromatin compaction are based on a single chromatin mark (H3K9me2,3) and one example image of H2B-GFP, respectively. To truly understand the consequence of HP1alpha loss they authors would have preferentially performed chromatin accessibility analyses such as ATAC seq and/or super resolution microscopy.

2. Similarly, conclusions of HP1alpha being essential for nuclear shape maintenance (Figure 3) were made based on just 25-30 nuclei – given the very elegant and efficient Auxin degron system and simple 2D model system, it is unclear why so few cells were analyzed. In additional, instead of expressing shape abnormalities by a descriptive classification (% of cells with abnormal nuclei) the authors should include a panel of detailed parameters to precisely quantify measure cell shape, such as solidity and the EFC ratio of nuclear shape (which some of the co-authors established in a recent publication).

3. Evidence for chromatin-chromatin crosslinks being affected in the I165E mutants should be experimentally demonstrated for example using super resolution

4. Mechanical measurements should be validated with an independent, higher throughput mechanical manipulation method, such as AFM or imaging-based methods

5. The validity of the polymer model that excludes a role for HP1-lamina or inner nuclear membrane interactions in nuclear mechanics is based solely on the short- vs long-extension stiffness data that the authors have previously validated for chromatin vs lamina-dependent mechanical regimes. The role of HP1 and lamina interactions through LBR or PRR14 should be experimentally excluded with mutants or LBR/PRR14 siRNA, in particular as these proteins have been shown to be important for nuclear shape maintenance.

6. Studies around the role of HP1alpha in mitotic segregation are not described clearly and the connection with chromosome mechanics remain unclear. The experiments are severely underpowered with only very few cells analyzed for mechanical measurements (just 8-14 nuclei) and 3-fold differences in sample size per condition (Figure 5G" "n=15-50 mitotic cells for each condition…") for metaphase misalignment measurements.

*Reviewer #2:*

The manuscript from Strom, Biggs, and Banigan, et al. is an outstanding effort and an important contribution to understanding the complex roles chromatin and protein effectors play in the nucleus. It certainly rises to the level of publication, and I found it to be generally very exciting and compelling work. I have one major criticism of the work below, but would be happy to accept the paper for publication.

The major criticism I have of the work is the validation of the HP1 protein after addition of the degron and sfGFP. Especially, given that there are a number of findings in this work are seemingly contrary to expectations, i.e no change in gene expression or H3K9 maintenance. From my reading, the only validation in the manuscript that I see that adding the degron and GFP onto the C-terminus of HP1a (more than tripling the MW right next to the protein-protein interaction hub of HP1) does not affect its biology is a localization measurement. And while I realize that it is not easy to validate a structural protein, it would be nice to have some more corroboration or comparison to experiments that have investigated the formation of heterochromatin by HP1 proteins.

For example, to reconcile these data with previous findings, one might predict that gene expression or H3K9 levels should eventually recapitulate previous knockdown/out studies if the auxin treatment were to persist, provided cell type and cell cycle considerations. While RNA seq experiments are fairly involved, I would think it possible to test the methylation levels in the auxin-HP1 line after a few days of depletion to see if levels do come down to previously reported levels, at least at specific satellite sequences. Alternatively, it might be easier to assay the AID-GFP-HP1 at exogenous sites as has been previously reported. Though, it is noted however that the dimerization mutant does perform as expected.

In addition to validation, it would also be helpful to more directly compare the timing of your experiments with those done elsewhere to give the reader a better idea of when changes (epigenetic, transcriptional) are likely to arise, etc. Though, discussion is somewhat confounded by the diversity of HP1 activity in different organisms. There is the distinction of >24 hours in the discussion, but there are several HP1 experiments that show faster changes than this, specifically compaction following recruitment and transcriptional repression at exogenous sites.

*Reviewer #3:*

The authors use state-of-the-art genome engineering and degron tagging approaches combined with transcriptional profiling and sophisticated biophysical measurements. Most of the results are clear and convincing, but the section on mitotic chromosomes has substantial weaknesses and should either be removed or extended to substantiate the key conclusions. The main part on HP1alpha's role in interphase nuclei, however, is interesting and well supported by the data.

1. Localization of HP1alpha on mitotic chromosomes. The authors claim that HP1alpha remains bound to mitotic chromosomes, but this is not visible in the data shown in the paper. Prior studies have in fact clearly shown that HP1alpha completely dissociates from chromosome arms, while only a small fraction remains bound to pericentric chromatin (Hirota et al., 2005; Fischle et al., 2005; Serrano et al., 2009; Akram et al., 2018). The authors reference these papers to provide a rationale for studying HP1alpha's potential role in mechanical rigidity of chromosome arms, but they neglect the fact that the previous studies showed that HP1alpha only localizes to a very small region surrounding centromeres. This very limited enrichment of HP1alpha at pericentromeric regions is inconsistent with a direct major mechanical role of HP1alpha in stiffening chromosome arms as the authors suggest.

Figure 5A shows HP1alpha-GFP fluorescence throughout the entire cell with a potentially very slight enrichment in the central region. Without counterstaining DNA or chromatin, it is not possible to assess whether any HP1alpha remains bound to chromosomes, as chromosome positions are not known. The authors should provide high-resolution figures of HP1alpha-GFP counterstained with a DNA dye, ideally in live cells. They should further image chromosome spreads as in the prior work by Serrano et al., 2009 to clarify where on chromosomes HP1alpha binds. Figure 5C shows a fuzzy fluorescence signal for HP1alpha, but the image quality is insufficient to assess whether this is on chromosome arms or on pericentromeric regions. Moreover, without a control cell that does not express GFP-tagged HP1alpha, it is not possible to assess to which extent the fuzzy fluorescence signal shown in Figure 5C represents background autofluorescence. Alternatively, as the mitotic chromosome stiffness measurements are also not very convincing (see next point), the authors might consider removing the entire section on mitotic chromosomes.

2. HP1alpha's contribution to mechanical rigidity of mitotic chromosomes. The authors claim that HP1alpha depletion "strongly impairs" mitotic chromosome mechanics. Their microneedle-based force extension experiments shown in Figure 5E, however, show a large variability between individual measurements but only a very slight difference between HP1alpha-AID-sfGFP cells treated with auxin compared to untreated cells. There is a star that might indicate statistical significance (this is not explained in the legend), but even if that were true then the effect size would still be so small that biological relevance is questionable. Given the concerns about HP1alpha localization, the overall evidence supporting a "major role" of HP1alpha in the stiffness of mitotic chromosomes is insufficient to support this conclusion. The authors might either perform additional experiments to substantiate the conclusions or remove the part on mitotic chromosomes.

3. HP1alpha's role in regulating nuclear shape. The authors suggest that the softening of interphase nuclei resulting from degradation of HP1alpha directly causes aberrant nuclear morphologies. However, they also show that depletion of HP1alpha leads to chromosome missegregation, which is known to lead to aberrant nuclear morphologies owing to perturbed nuclear assembly during mitotic exit. To assess to which extent the nuclear morphology defects in HP1alpha depleted cells are caused directly via nuclear softening or indirectly via chromosome missegregation, the authors should separately analyze nuclear morphology shape changes as shown in Supplementary Figure 2 for those cells that have passed through mitosis after HP1alpha degradation versus those cells that remained in interphase during the entire imaging duration.

---

## [Author Response]

Essential Revisions:1. The section of the manuscript focused on effects on mitosis is weak and should be removed or substantially improved as outlined in detail in the reviews.

In the revised manuscript, we have substantially improved the section of the manuscript focused on the role of HP1α in mitotic chromosomes and function by adding experiments to address the Reviewers’ critiques. Specifically, we have: (A) Increased the number of micromanipulation force measurements to improve the statistics; (B) Imaged HP1α-AID-sfGFP on isolated chromosome bundles (in addition to the original imaging in cells) to assay its presence on chromosome arms and pericentromeric regions; (C) Imaged HP1α-AID-sfGFP in live cells, in which we previously observed HP1α-AID-sfGFP at pericentromeric regions; (D) Provided more measurements of metaphase and anaphase fidelity with and without auxin treatment; (E) Tracked the effects of mitotic errors on subsequent interphase nuclear shapes. Further details are provided below. Overall, our new data strengthens our novel findings of the role of HP1α in maintaining mitotic function and fidelity.

A. We have performed more single isolated chromosome micromanipulation force measurements, as requested by the reviewers (6-8 new measurements each, bringing the total to 14-20 each; Figure 5E). Our new data further supports our initial conclusions that HP1α is a mechanical component of mitotic chromosomes, as its depletion decreases doubling force from

262 +/- 49 to 148 +/- 12 pN/μm (p = 0.03). Furthermore, the new measurements have improved the statistics of the measurements for chromosomes from auxin+methylstat-treated cells, which remain similar to methylstat-treated chromosomes (p = 0.17), but are significantly different from those with auxin treatment (p = 0.04). This data supports the conclusion that levels of histone methylation are dominant in mitotic chromosome mechanics as compared to HP1α. Interestingly, this finding differs from observations in interphase nuclei, where HP1α and histone methylation appear to contribute similarly to nuclear stiffness. Thus, the measurement of mitotic chromosome stiffness shows that some mechanical principles are the same as for interphase nuclear mechanics (HP1α is a mechanical element) and others are different (mitotic chromosome stiffness is dominated by histone methylation rather than having similar contributions from both HP1α and histone methylation).

B. We have added new data quantifying the level of HP1α on isolated mitotic chromosome bundles. In the original manuscript we reported levels of HP1α-AID-sfGFP fluorescence in live mitotic cells, which after 4-hour auxin treatment decreased on average by 89% (now Figure 5B). We have added fluorescence measurements of ex vivo isolated chromosome bundles (Figure 5B, C, and figure supplement 1). We report that bulk fluorescence indicates that HP1α-AID-sfGFP associates with the chromosome bundle, and it decreases by ~75% after 4-hour auxin degradation. We have now provided multiple examples showing HP1α-AID-sfGFP fluorescence on isolated chromosome bundles. Line scans reveal that HP1α-AID-sfGFP fluorescence is present on chromosome arms (Figure 5C, with more examples in Figure 5—figure supplement 1) as well as at pericentromeric foci, as previously reported. The novelty of this finding is that HP1α on the chromosome arms may only be detectable by imaging isolated mitotic chromosome bundles because isolation removes chromosomes from the high background fluorescence in the mitotic cell. Below, in point C we note that pericentromeric foci are detectable by confocal, in agreement with the previously published reports cited by Reviewer 3, and we have now added HP1α-AID-sfGFP and Hoechst-stained DNA images to the manuscript (Figure 5—figure supplement 1C). We also now note that HP1α on chromosomal arms and at the pericentromere both should be able to contribute to mitotic chromosome mechanics. Altogether, this data strengthens our previous conclusion that HP1α is present on mitotic chromosomes and extends it to include the chromosome arms.

C. We have added new data imaging HP1α-AID-sfGFP in living cells via confocal microscopy. This data shows HP1α at the centromere relative to DNA/chromatin staining/fluorescence (see Figure 5—figure supplement 1C), which is in agreement with our own isolated bundle images as well as many other published reports.

D. To further bolster the mitosis section, we also increased the number of events measured for abnormal metaphase alignment and anaphase/telophase segregation. We have included another experimental replicate of – auxin and + auxin at 16 hours to increase the total number of events. This new data agrees with our original measurements. We have also included three experimental replicates measuring metaphase and anaphase/telophase abnormal events after rapid 4-hours of auxin-induced HP1α degradation (Figure 5F). These measurements further verify that loss of HP1α results in more metaphase misalignment and anaphase/telophase missegregation. Furthermore, it reveals that longer-term HP1α loss results in more mitotic failures, as we observe a significant increase in abnormal events when comparing short-term degradation of HP1α (4 hour auxin treatment) vs. longer-term degradation (16 hour).

E. In response to comments by Reviewers 1 and 3, we have provided new data analyzing how mitotic abnormalities in HP1α degraded cells lead to abnormal interphase nuclear morphology.

i. In the original manuscript, by tracking 42 cells' nuclear curvature measurements over 1 to 12 hours after addition of auxin, we clearly showed that abnormalities in nuclear morphology emerge during interphase (see Figure 2—figure supplement 2). Thus, over the short time intervals of rapid HP1α degradation, it appears that abnormal nuclear morphology can arise, independent of mitosis.

ii. Our new data provide time-lapse imaging to reveal that control untreated cells successfully complete mitosis and produce normally shaped sister nuclei. However, after 24 hours of auxin treatment, nearly half of HP1α-degraded nuclei tracked through mitosis exhibit abnormal mitotic events (in agreement with Figure 5F). However, HP1α degraded cells resulting in normal vs abnormal mitosis both produce sister nuclei with higher nuclear curvature(Figure 5G). Thus, abnormal mitosis is not required to produce abnormally shaped/high curvature sister nuclei post mitosis. Overall, abnormal nuclear morphology can arise due to the loss of nuclear shape regulation by HP1α during interphase (over short timescales) or arise after mitosis resulting from HP1α degradation (over long timescales).

As we discuss in the more detailed reply below, our reporting of experimental replicates and the associated number of measured events was confusing in the original manuscript. This may be why Reviewer 1 (and possibly others) was unsure about how many actual measurements were taken. We have clarified this in the figure legends and we now list the number of measurements for each of the replicates. We have also clearly defined asterisks denoting statistical significance. We cover this point again in Editor point 5.

2. Cell shape analyses should show actual measurement data (not phenotypic classification) and the dependency of the nuclear shape defect on mitosis should be addressed.

The original manuscript contained data addressing this point. In the revised manuscript, we better highlight this original data, state the quantified data instead using a phenotypic term, and provide increased numbers of nuclei for these measurements and new experiments to further address these points. The data quantify nuclear shape by either shape solidity (ratio of area to convex area) or nuclear curvature (phenotypic classifications of nuclei were then made based on these measurable quantities). The data shows that HP1α is essential for maintaining nuclear shape. The data also reveals that HP1α degradation decreased nuclear shape solidity or increased nuclear curvature. Higher solidity and decreased curvature can be recovered by increasing levels of histone methylation via methylstat or by introducing exogenous wild type HP1α, but not by exogenous expression of the dimer mutant HP1α^I165E^. This reveals that overall H3K9 methylation levels function independently of HP1α for these functions over 4-16 hour timescales, and that HP1α’s nuclear shape maintenance function depends on HP1α dimerization.

More specifically, the original manuscript measured nuclear shape changes via (A) nuclear shape solidity (Figure 2G); (B) timelapse nuclear curvature measurements (Figure 2—figure supplement 2); and (C) population nuclear curvature measurements, which we provide new data increasing biological replicates (Figure 3). We have revised the manuscript to include average nuclear shape solidity measurements in the text section “HP1α is a major mechanical component of the interphase nucleus that contributes to nuclear shape maintenance” and Figure 2 figure legend. Furthermore, we clarify we are reporting an increase in the number of nuclei with low solidity levels (< 0.96), which we for reference refer to as abnormal or irregular. To further address the reviewers’ concerns, we now additionally include: (D) new data, essentially recapitulating the experimental measurements of solidity and finding the same results, significance, and conclusions as already included in Figure 2; (E) new data to address the effect of abnormal mitotic events on nuclear morphology, as requested by the reviewers. Details are listed below.

A. Nuclear shape solidity. Original data and new/more data provided – Figure 2G consists of: “3 experiments each (shown as black dots) each consisting of n = 109, 102,105 control ; n = 137, 115, 165 auxin, n = 31, 34, 32 methylstat, and n = 102, 92, 78 auxin methylstat”. In the figure legend we have added, as requested by the reviewers, that average solidity also has similar statistical significance: “ Average measurements were similar for control, methystat, and auxin with methylstat 0.971+/- 0.0001 but different for + auxin 0.969+/-0.0015, p=0.005.” The standard deviation increased over ten-fold in + auxin conditions signalling a drastic change in the distribution of solidity. Specifically, under these conditions, there is an increase in the number of nuclei with low solidity (< 0.96 solidity as shown in Figure 2G). Thus we graphed the % of nuclei with low solidity to show that nuclear shape had destabilized.

B. Time series of nuclear curvature. Original data – We tracked nuclear curvature for single interphase nuclei over time during auxin-induced HP1α degradation, which clearly shows loss of shape was independent of mitosis and coincident with HP1α loss (Figure 2—figure supplement 2). Here we do measurements of nuclear curvature and assay for the presence of HP1α-AID-sfGFP each hour for 12 hours for 42 individual nuclei of – or + auxin. This data clearly shows that + auxin results in increased nuclear curvature coincident with HP1α-AID-sfGFP degradation during interphase and independent of mitosis. These experiments provide actual measurement data and address the dependency of the nuclear shape defects on mitosis (also see point E below).

C. Population nuclear curvature measurements. New data and original data – In the revised manuscript we provide 5 new experimental replicates to go along with the 3 original experimental replicates (8 total). This data strongly supports the major conclusion of the paper that HP1α dictates nuclear morphology though its dimerization. Population nuclear curvature measurements of HP1α degradation and rescue with exogenous HP1α (Figure 3) recapitulate both of these major findings from Figure 2G and Figure 2 – supplemental figure 2. In this figure, we now clearly state that for this data: “ Eight experimental replicates were measured for each condition (denoted as black dots) consisting – auxin, n = 37, 45, 45,57, 58, 57, 55, 54; + auxin, n = 60, 44, 41, 60, 60, 60, 60, 60 ; + auxin and WT exogenous rescue, n = 27, 30, 36, 60, 60, 60, 60, 19; + auxin and I165E exogenous rescue, n = 38, 40, 50, 59, 56, 58, 58, 51. P values reported as n.s > 0.05, * < 0.05, **< 0.01, ***< 0.001, ****<0.0001.”

D. New experiments recapitulating previous results of Nuclear Shape Solidity. New data shown here in Author response image 1. As a further proof of the validity of our data, we have redone the solidity measurements in a different lab setting to increase the number of measurements and show the data can be recapitulated. Here, we show that average solidity and % of nuclei with a solidity less than 0.96 are similar to our original data. The slight change in absolute numbers is due to different microscopes (widefield vs. confocal). 3 experiments each (shown as dots), with each experiment consisting of n = 96, 96, 97 untreated control (unt); n = 109, 96, 115 auxin (aux); n = 47, 48, 49 methylstat (ms); and n = 87,101,79 auxin methylstat (aux ms). The average nuclear shape solidity of auxin-treated cells is statistically significantly different (p < 0.05), while the rest are statistically similar (0.955 auxin vs 0.959, 0.960, 0.961 +/- 0.001 for control, methylstat, and auxin + methylstat, respectively). While these absolute measurements of solidity are less than in the original manuscript, the % abnormally shaped nuclei (solidity < 0.96) increases similarly by ~10% for auxin-treated nuclei and the change is significant (p < 0.05). The aim of sharing this data here is to show that these results are reproducible, robust, and rigorous through multiple measurements, measurement types, modalities, and replicates.

**Author response image 1. sa2fig1:** 

E. Abnormal mitotic events. New data – To directly address the question of the dependency of the nuclear shape defects on mitosis we have included new data in the manuscript. From point B above, we clearly show in the original manuscript that abnormal nuclear shape is independent of mitosis. To determine the effect of abnormal mitosis (originally reported in Figure 5), we tracked nuclei/cells through mitosis either – auxin or + auxin for 24 hours. We now report that both normal and abnormal mitosis in +auxin conditions results in daughter nuclei with higher nuclear curvature measurements than – auxin (Figure 5 G). This new data provides the interesting insight that abnormal nuclear morphology can be independent of mitosis after rapid (4 hours) HP1α degradation, but HP1α degradation results in higher nuclear curvature in sister nuclei post mitosis on longer time scales (> 24 hours).

3. The role of HP1-lamin interaction in nuclear mechanics should be addressed.

The revised manuscript addresses the possible role of HP1α in linking chromatin to lamins/lamina or otherwise interacting with lamins through new experiments that investigate HP1α’s role in maintaining peripheral H3K9me^2,3^ heterochromatin. We note that the experiments requested by Reviewer 1 (modulating LBR and PRR14) are a significant undertaking that would not directly address the main points of the manuscript. As we describe below, our new experiment more directly addresses the possibility of HP1α as a chromatin-lamina linker and/or lamin interactor.

First, in the original manuscript Figure 4, computational simulations showed that altering the level of chromatin-chromatin crosslinking changes nuclear mechanics differently as compared to altering the amount of chromatin-lamin linkers. These simulations test the possible mechanical roles of HP1α as either a chromatin-chromatin crosslinker or chromatin-lamin linker. The simulations show that loss of chromatin crosslinkers decreases the short-extension spring constant while having little effect on the long-extension spring constant. Furthermore, the simulations show that the loss of chromatin-lamin linkers results in decreased nuclear spring constant for both the short and long extensions. The results for the simulations with different levels of chromatin crosslinking match experimental measurements of nuclear mechanics upon auxin-induced degradation of HP1α. Together, these results show that loss of HP1α perturbs nuclear mechanical response in a manner consistent with a decrease in crosslinking within the chromatin polymer gel, but inconsistent with changes to the frequency of mechanically stable chromatin-lamina links. Based on these results, combined with our HP1 dimerization experiments and HP1α’s established ability to bridge nucleosomes (Machida et al. *Mol Cell* 2018), we conclude that the likely primary mechanical function of HP1α is to crosslink chromatin.

Second, we have provided new data to address possible HP1α-lamin interactions. Reviewer 1 notes that there are known chromatin-lamin linkers, including LBR and PRR14. As shown by previous work (Dunlevy et al., 2020; Giannios et al., 2017; Nikolakaki et al., 2017; Poleshko et al., 2013; Solovei et al., 2013), depletion of these known chromatin-lamin interacting proteins results in a significant decrease in peripheral H3K9me^2,3^ heterochromatin. To test if HP1α has a role as a chromatin-lamina linker, we measured both the level of HP1α at the nuclear periphery and the density of H3K9me^2,3^ heterochromatin at the nuclear periphery before and after HP1α degradation. Known chromatin-lamin linkers PRR14 and LBR are both themselves enriched at the nuclear periphery AND upon loss of them the peripheral enrichment of H3K9me^2,3^ is significantly decreased (Poleshko et al., 2013; Solovei et al., 2013). Unlike H3K9me^2,3^, which is enriched at the periphery (1.5 average periphery/average internal signal), HP1α peripheral signal/average (0.86) is similar to DNA staining (0.99), with both showing no peripheral enrichment. Rather, both HP1α and DNA appear to be equally partitioned between the nuclear periphery and the nuclear interior (average periphery/interior ratio~1). Thus, in this cell line, HP1α does not exhibit peripheral enrichment that is characteristic of known chromatin-lamin tethering proteins such as PRR14 and LBR. Furthermore, upon degradation of HP1α, the peripheral enrichment of H3K9me^2,3^ does not change significantly. Thus, unlike known chromatin-lamin linkers LBR and PRR14, we find that HP1α does not have a major role in maintaining peripheral localization of H3K9me^2,3^ in this cell line. Taken together this new data supports our conclusion that the mechanical contributions of HP1α are not due to associated chromatin-lamin interactions.

4. Rev#2 makes the important point of validating the function of the engineered protein, which relates to concerns of Reviewer#1 on the quality (abundance) of the data showing surprising absence of effects on heterochromatin and transcription. This should be experimentally addressed or extensively discussed, as Rev#2 suggests.

To address this point, we have added new experiments validating the function of the engineered protein. We have also added more discussion of how the lack of changes in heterochromatin organization and gene transcription, while apparently surprising, is actually in agreement with previous studies of mammalian cells. We provide several key points and new experiments to support this assertion:

A. Firstly, we modify the endogenous HP1α loci rather than using exogenous HP1α, which can result in overexpression due to not being under the control of the endogenous promoter. Thus, our modified HP1α is under native promoter control.

B. We have provided new RNA-seq data showing that the transcription profile does not change significantly for the modified HP1α-AID-sfGFP cell line as compared to the parent cell line. RNA-seq reveals 76 upregulated and 56 downregulated transcripts, which represents just 0.8% of the >16,600 genes. GO analysis reveals no change to the nucleus or chromatin proteins. GO analysis of cellular function returns only extracellular changes for upregulated genes (matrix, region, space, exosome, and vesicle) and no significant GO term for downregulated genes. This new data is added as a Data Supplement 3.

C. The main measurables in the manuscript for HP1α function are similar for parental and the AID-sfGFP modified cell lines, while they differ significantly when HP1α is degraded via auxin. These measurables include: (1) general HP1α distribution, Figure 1 (2) H3K9me^2,3^ levels, Figure 2—figure supplement 1D (3) interphase nuclear mechanics, Figure 2 (4) nuclear shape, Figures 2 and 3 (5) mitotic fidelity, Figure 5—figure supplement 1D.

D. We note that C-terminal tagging of HP1α has been shown to not disrupt function through rescue experiments in previous studies (Dialynas et al. 2007 J Cell Sci) as well as in our studies. Here in Author response image 2, we also supply supporting images of endogenous subnuclear localization of HP1α in parental U2OS cells stained with anti-HP1α (A), which have similar localization to the nucleolar periphery as an exogenous C-terminally tagged construct (B). C-terminally tagged endogenous HP1α-GFP-AID also localizes to nucleolar periphery (C) and directly overlaps with both N-terminally tagged (D) and C-terminally tagged (E) exogenous mCherry constructs, suggesting that the presence and orientation of the tag does not disrupt HP1α chromatin binding or subnuclear localization. Also, exogenous wild-type HP1α rescued nuclear shape while the dimer mutant did not, which Reviewer 2 noted was a significant finding to support the functionality of the endogenous modified HP1α.

**Author response image 2. sa2fig2:** A. Antibody stain with anti-HP1α in parental U2OS showing endogenous localization to heterochromatic areas and around nucleoli. B. Antibody stain with anti-HP1α in parental U2OS expressing exogenous C-terminally tagged HP1α-mCherry, showing similar localization around nucleoli and in heterochromatic areas. C. Antibody stain with anti-HP1α in HP1 α-GFP-AID C-terminally tagged cell-line, showing colocalization between antibody and GFP tag, and normal localization around nucleoli. D. Co-expression of N-terminally tagged exogenous mCherry-HP1α and endogenously tagged HP1 α-GFP-AID showing no difference in protein localization between N-terminal and C-terminal tagged populations. E. Co-expression of C-terminally tagged exogenous HP1α-mCherry and endogenously tagged HP1α=GFP-AID showing no difference in protein localization between endogenous and exogenous C-terminally tagged proteins.

E. The seemingly surprising absence of effects of HP1α depletion can be understood by considering the now well documented differences between fly and mammalian HP1α. In particular, previous studies in *Drosophila* show that HP1α depletion can modify position effect variegation (Eissenberg et al. *Genetics* 1992) and modulate levels of transposon and satellite RNA expression (Sienski et al. *Cell* 2012) as well as cell cycle regulators (De Lucia et al. *Nucleic Acids Res* 2005). In murine cells, HP1α is involved in repression of major and minor satellite RNA (Eissenberg and Elgin *TrendsGenet* 2014) and also helps regulate olfactory receptor expression (Clowney et al. *Cell* 2012). However, as we note in the manuscript, published studies of HP1α depletion or loss in human cells are in agreement with our findings of lack of changes in transcription and histone methylation (Zeng et al. *Epigenetics* 2010) (Maksakova et al., 2011), but a significant change in mitotic fidelity(Levine, Vander Wende, and Malik 2015; Abe et al. 2016) Furthermore, studies finding that HP1α acts as a repressor of transcription typically focused on a single or small set of sites to which HP1α was highly recruited (Erdel et al. *Mol. Cell* 2020; Li et al. *Development* 2013).

5. There is concern about the small sample sizes and the differences in sample size between experimental groups of individual experiments. This concern should be addressed by increasing the sample size and making the sample sets being compared similar in size.

We have revised the manuscript to better communicate the number of replicate experiments and the number of measurements for each experiment. We have also added new data to increase our sample sizes where requested, especially to address the above Editorial points 1, 2, and 3. The two most important things to note are that none of the data changed significance upon increasing n’s and we have, in most places, doubled the quantity of data by adding new experiments. Below we detail the sample sizes for the specified data.

Figure 1 – unchanged, but new RNAseq data set

i. Revised reporting of Figure 1—figure supplement 1 shows measurements for 3 replicate experiments with 20, 20, 20 nuclei each.

ii. RNAseq data supplements for parental vs. HP1α-AID-sfGFP modified (new addition), HP1α-AID-sfGFP untreated vs. auxin 4 hours, and HP1α-AID-sfGFP untreated vs. auxin 16 hours

Figure 2 – revised sample size reporting for clarity and provided new related data within this response

i. Panels F (protein levels) and G (nuclear shape solidity) updated in figure legend: “3 experiments each (shown as black dots), each consisting of n = 109, 102, 105 control ; n = 137, 115, 165 auxin, n = 31, 34, 32 methylstat, and n = 102, 92, 78 auxin methylstat”

ii. New supporting data in the reply (see above Editor point 2)- Nuclear shape solidity was measured again in new experiments and we observed the same outcome; please see data above in response to Editor point 2 D: “3 experiments each (shown as dots) each consisting of n = 96, 96, 97 control ; n = 109, 96, 115 auxin, n = 47, 48, 49 methylstat, and n = 87,101,79 auxin methylstat. Auxin is statistically significantly different p < 0.05 while the rest are statistically similar. While these absolute measurements of solidity are less than in the original manuscript, % abnormally shaped nuclei (solidity < 0.96) increases similarly by ~10%.”

iii. New data Figure 2—figure supplement 1 – “Normalized to parental control, immunofluorescence signal of H3K9me2,3 for parental and HP1α-AID-sfGFP modified cell lines without or with treatment of auxin for 16 hours and/or methylstat for 48 hours. Three experimental replicates parental -/- n = 216, 128, 229; +/- n = 307, 324, 215; -/+ n = 188, 115, 155; +/+ n = 184, 258, 284; HP1α-AID-sfGFP -/- n = 108, 101, 104; +/- n = 136, 114, 164; -/+ n = 30, 33, 31; +/+ n = 102, 91, 77. (E) Normalized immunofluorescence for H3K9me2,3 and H3K9ac in HP1α-AID-sfGFP – or + auxin for 4 days (96 hours) for six experimental replicates – auxin (218, 246, 263, 186, 238, 265) and + auxin (188, 184, 174, 208, 199, 187). P values reported as no asterisk > 0.05, * < 0.05, **< 0.01, ***< 0.001.”

Figure 3 – New data to increase sample size in all cases (5 new experimental replicates were added to the 3 original experimental replicates)

i. Nuclear curvature measurements: “Eight experimental replicates were measured for each condition (denoted as black dots) consisting of – auxin, n = 37, 45, 45, 57, 58, 57, 55, 54; + auxin, n = 60, 44, 41, 60, 60, 60, 60, 60 ; + auxin and WT exogenous rescue, n = 27, 30, 36, 60, 60, 60, 60, 19; + auxin and I165E exogenous rescue, n = 38, 40, 50, 59, 56, 58, 58, 51. P values reported as n.s. > 0.05, * < 0.05, **< 0.01, ***< 0.001, ****<0.0001.”

ii. New data in the reply – parental nuclear curvature measurements are similar (- or + auxin, 5 replicate experiments) to untreated HP1α-AID-sfGFP modified. See Reviewer 2 major point 1. Also added to the revised manuscript in the main text.

Figure 4

i. Additional simulations have been performed to improve the statistics. We have considered Reviewer 2’s comment in reporting the number of simulations (further addressed below) and clarified the wording.

ii. New experimental data on the fraction and enrichment of H3K9me^2,3^ and HP1α at the nuclear periphery: “One experiment measuring – auxin n = 93 and + auxin n = 149.”

Figure 5 – new data to increase sample sizes in all cases

i. Panels B (measured HP1α in isolated chromosome bundles) and E (micromanipulation force measurements). In general, we nearly doubled sample size by adding 6-8 measurements for each : “For B-E n = 20 for control and auxin treated, n = 16 for methylstat and n = 14 auxin methylstat treated.”

ii. Panel F abnormal metaphase and ana/telophase events, of which we add one new experiment for control and 16 hours auxin while we added a new condition – 4 hours auxin – with three new experimental replicates : “Metaphase misalignment 3-4 experiments (black dots) consisting of n = 16, 15, 20, 37 -aux, n = 33, 33, 24 +aux 4 hours, n = 22, 48, 58, 54 +aux 16 hours. Anaphase missegregation 3-4 experiments (black dots) consisting of n = 29, 23, 30, 30 -aux, n = 32, 29, 18 +aux 4 hours, n = 20, 35, 36, 45 +aux 16 hours.”

iii. All new data, Panel G abnormal mitoses and nuclear curvature for sister nuclei. “(G) HP1αAID-sfGFP cells – auxin or + auxin for 24 hours were tracked through mitosis to determine if abnormal mitosis results in abnormally shaped daughter nuclei measured via nuclear curvature (parental 34 nuclei from 17 mitoses; – auxin 46 nuclei from 23 mitoses; + auxin 51 nuclei from 26 mitoses, p = 0.00001). Percentage of abnormal mitosis presented in Figure 5—figure supplement 1D.”

Reviewer #1:Strom and co-authors describe a new, very intriguing role for HP1alpha in chromatin mechanics and nuclear shape maintenance which is independent of known nuclear mechanoregulators: histone methylation or nuclear lamina, and which is instead driven by HP1alpha-mediated crosslinking of chromatin. While conceptually exciting, the central conclusions lack definitive experimental evidence. Further, key measurements lack true statistical power and the polymer model predictions are not stringently challenged. Finally, the functional relevance of this HP1-mediated mechanical regulation remains mostly unexplored, as the chromatin mis-segregation phenotype is only superficially analyzed and its connection with chromatin mechanics is elusive.1. Exclusion of contribution of chromatin methylation and lack of changes in chromatin compaction are based on a single chromatin mark (H3K9me2,3) and one example image of H2B-GFP, respectively. To truly understand the consequence of HP1alpha loss they authors would have preferentially performed chromatin accessibility analyses such as ATAC seq and/or super resolution microscopy.

The Reviewer raises concerns over (1) the use of one methylated histone mark and (2) the use of H2B and (3) they ask for further data to support the result that chromatin compaction does not change. In the original manuscript we look at (1) H3K9me^2,3^ because it is the histone mark that HP1α binds and is thus the most relevant and (2) H2B because it is widely used to measure chromatin density due to its prevalence throughout chromatin. (3) We have revised the manuscript to remove any mention of changes in global chromatin compaction, which was not measured and is not necessary for the main conclusions of the paper.

1. The conclusions we make in the manuscript are supported by the data provided. In the original manuscript, we assayed changes in the most relevant constitutive heterochromatin maker, H3K9me^2,3^, which HP1α binds, and we found no change upon HP1α degradation via auxin treatment for 4 hours (Figure 2, E and F). In the revised manuscript, we include new data that auxin-induced degradation of HP1α does not change H3K9me^2,3^ histone methylation over 16 hours or 96 hours (Figure 2—figure supplement 1, E and E). The 4 day (96 hour) auxin-induced degradation also shows no change in euchromatin marker H3K9ac.

2. The Reviewer is incorrect that “one example image of H2B-GFP” is used to conclude that there is no change in chromatin compaction. In the original manuscript, we measured colocalization of dense H2B-miRFP and HP1α-sfGFP-AID, which showed that histonedense foci remained upon degradation of HP1α in three experimental replicates of 20 nuclei each (60 total). We have clarified these points in the revised manuscript, specifically stating that local chromatin compaction is not altered upon HP1α loss.

3. The Reviewer suggested that chromatin compaction and accessibility could be further investigated through ATAC-seq or super-resolution. Indeed, these would be interesting experiments, but they would also represent significant undertakings that would not directly support the central conclusions of this manuscript, which primarily concern the architectural role of HP1α in nuclear organization and function. We have revised the manuscript to remove any statement suggesting global changes in chromatin compaction, as we specifically measure (1) relevant histone methylation and (2) local chromatin foci compaction.

To further measure heterochromatin H3K9me^2,3^ we have added new data and analysis of enrichment at the periphery of the nucleus in 93-149 cells. This analysis shows that HP1α degradation does not change H3K9me^2,3^ heterochromatin enrichment at the periphery or DNA partitioning at the periphery (New Data Figure 4 E and F). Taken together, our data reveals that on the measured timescales, HP1α has no significant role in changing the level of its binding target H3K9me^2,3^ methylation or the euchromatic marker H3K9ac, or the enrichment of its binding target H3K9me^2,3^ at the periphery, or local compaction of chromatin.

2. Similarly, conclusions of HP1alpha being essential for nuclear shape maintenance (Figure 3) were made based on just 25-30 nuclei – given the very elegant and efficient Auxin degron system and simple 2D model system, it is unclear why so few cells were analyzed. In additional, instead of expressing shape abnormalities by a descriptive classification (% of cells with abnormal nuclei) the authors should include a panel of detailed parameters to precisely quantify measure cell shape, such as solidity and the EFC ratio of nuclear shape (which some of the co-authors established in a recent publication).

The Reviewer requested (i) more data and (ii) specific measurements. (i) First, we have more than doubled the data for Figure 3 (nuclear curvature) with 5 new experimental replicates. In addition, the Figure 2 measurements were completely recapitulated in new experiments, as shown in Editor point 2D above. (ii) Figure 3B and Figure 2—figure supplement 2 report nuclear curvature, and Figure 2 measures nuclear shape solidity. We have revised the manuscript to include average nuclear shape solidity reported in the text, which supports the major conclusions of the manuscript.

The original manuscript's reporting of the number of nuclei was confusing to multiple reviewers. While it may have been unclear, we measured > 25 nuclei for each of 3 experimental replicates for nuclear curvature. The fewest nuclei were measured in the original Figure 3, which still reported measurements of 83 nuclei (27, 30, 26 nuclei for each replicate). In the revised manuscript we have significantly increased sample size for nuclear curvature measurements (now Figure 3 has 8 experimental replicates and Figure 5G measures nuclear curvature post normal and abnormal mitosis). We have revised the manuscript to more clearly indicate the total number of cells measured for each replicated.

Below we share how we now clarify and fully report n for all nuclear shape measurements in the revised manuscript to address (i) quantity of data and (ii) how we quantify the data. The revised manuscript now clearly communicates the quantitative measurements we used to determine maintenance or loss of nuclear shape, instead of using phenotypic wording. Specifically, we have revised the manuscript and figures to replace “abnormal/irregular” with specific measurements of (A) nuclear curvature, (B) individual nucleus curvatures over time, (C) and solidity.

A. New data has been added to Figure 3 which measures nuclear curvature. The revised manuscript now includes 8 biological replicates (3 original and 5 newly added) and states:

“Eight experimental replicates were measured for each condition (denoted as black dots) consisting – auxin, n = 37, 45, 45,57, 58, 57, 55, 54; + auxin, n = 60, 44, 41, 60, 60, 60, 60, 60 ; + auxin and WT exogenous rescue, n = 27, 30, 36, 60, 60, 60, 60, 19; + auxin and I165E exogenous rescue, n = 38, 40, 50, 59, 56, 58, 58, 51. P values reported as n.s. > 0.05, * < 0.05, **< 0.01, ***< 0.001, ****<0.0001.”

B. Figure 2—figure supplement 2 tracks nuclear curvature for 42 nuclei every hour over 12 hours for each untreated control and auxin-induced HP1α degradation. We would like to point out that this is a rigorous quantitative measurement of nuclear shape, and it is taken during HP1α-AID-sfGFP degradation and entirely during interphase (independent of mitosis). This data strongly supports the data in Figure 3 and provides a measure of time. This time measurement, along with the population average of nuclear curvature, shows a significant increase at the time point 5 hours after auxin addition, which coincides clearly with the time interval of HP1α-AID-sfGFP degradation (Figure 1 and 2).

C. New data panel Figure 5G measures nuclear curvature after mitosis for parental, HP1α- AID-sfGFP without auxin, and HP1α- AID-sfGFP with auxin. This data further confirms that sister nuclei after mitosis have higher nuclear curvature upon HP1α degradation.

D. Figure 2 solidity < 0.96 is used to determine abnormally shaped nuclei. The revised manuscript includes clear statements of number of measurements and now includes the average solidity that follows the same trends of significance: “ 3 experiments each (shown as black dots) each consisting of n = 109, 102,105 control ; n = 137, 115, 165 auxin, n = 31, 34, 32 methylstat, and n = 102, 92, 78 auxin methylstat. Average measurements were similar for control, methystat, and auxin with methylstat 0.971+/- 0.0001 but different for auxin 0.969+/-0.0015, p=0.005.”

We would also like to point out that we redid the solidity measurements and observed a similar decrease in average solidity with auxin-induced HP1α degradation relative to the other conditions as well as increase in number of nuclei with solidity less than 0.96 (see Editorial point 2D above).

Overall, we have provided new data and clarified the reporting of our quantitative measurements. Taken together, this data clearly shows loss of nuclear morphology upon degradation of HP1α that can be rescued by increased histone methylation (methylstat) or exogenous HP1α rescue, but not the dimer mutant HP1α I165E.

3. Evidence for chromatin-chromatin crosslinks being affected in the I165E mutants should be experimentally demonstrated for example using super resolution

The Reviewer requested new validation of the loss of chromatin-chromatin crosslinking via dimerization disruption in the HP1α I165E mutant. The HP1α I165E mutant has already been reported to lose its ability to dimerize; indeed single point mutations to the chromoshadow domain can disrupt dimerization (Brasher et al. EMBO J 2000, Lechner et al. Mol. Cell Biol. 2000, Thiru et al. EMBO J 2004, Lechner et al. Biochem Biophys Res Commun 2005). Additionally, human HP1α promotes intra- and inter-strand crosslinking of nucleosome arrays in vitro, and this ability is abolished by disrupting dimerization (Azzaz et al. 2014 J Bio Chem). Moreover, this mutant is widely used in the literature and shown by Number and Brightness analysis to exist as a monomer in living mammalian cells (Hinde et al., Sci Reports 2015; Machida et al., Cell 2018). Because this mutation has already been well established to disrupt dimerization of HP1α by other publications, we believe further validation is an unnecessary technical challenge. Super-resolution light microscopy techniques do not have a high enough resolution to directly visualize chromatin-chromatin crosslinks or their subsequent loss upon I165E dimerization. However, confirmation of HP1α as a di-nucleosome crosslinker has been accomplished with single particle electron microscopy (Machida et al., 2018 Cell), and recent methodological developments have allowed visualization of chromatin regions and found increased mesh density in heterochromatic as compared to euchromatic areas (Ou et al. Science 2017).

4. Mechanical measurements should be validated with an independent, higher throughput mechanical manipulation method, such as AFM or imaging-based methods

The Reviewer requests new validation of the force measurements. We note that micromanipulation is a well established technique for measuring the mechanical properties of both mitotic chromosomes and cell nuclei. Micromanipulation of mitotic chromosomes via micropipettes has been used to directly probe chromosome elasticity for over two decades (Houchmandzadeh et al. J Cell Biol 1997, Poirier et al. Mol Biol Cell 2000). Whole-nucleus micromanipulation was recently established by some of us (Stephens et al. Mol Biol Cell 2017), and the measurements were further validated by subsequent experiments, in part, by one of us (Hobson et al. Mol Biol Cell 2020) and another group (Shimamoto et al. Mol Biol Cell 2017). Therefore, we do not believe an independent validation is necessary.

Furthermore, we note a secondary, validating technique is extraneous for two additional reasons: (A) Micromanipulation force measurements provide unique insights into the identified two regimes of nuclear mechanical response, which are central to this paper and which standard AFM cannot provide; and (B) Micromanipulation force measurements have several internal experimental controls, which we describe below. This further bolsters confidence in the measurements included in the original manuscript, which show a clear statistically significant change in the chromatin dominated nuclear spring constant for HP1α degradation as compared to controls.

A. Micromanipulation has particular advantages over other techniques, such as AFM. In particular, standard AFM compression measurements of cell nuclei are incapable of separating the chromatin- and lamin-based force response regimes of the nucleus. In addition, standard AFM has to model force-compression as a Hertzian spring compressing an immovable substrate, and thus it lacks the ability to track both deformation and force accurately. In contrast, our micromanipulation extensional force measurement technique does not rely on the coarse estimations of AFM, but it instead allows unhindered, highly controlled, and fine tracking of extension vs. force. This allows micromanipulation to separate these mechanical regimes and observe that chromatin controls short extension nuclear force response while lamin A controls strain stiffening at longer extensions (> 3 μm, Stephens et al., 2017 MBoC). The technique and its novel findings were recently confirmed via a new more advanced Single-Plane Illumination Microscopy AFM technique (Hobson et al., 2020 MBoC). This first-of-its-kind combined SPIM AFM provides the needed fine measurements of deformation vs. force response necessary to separate the two force response regimes, further proving that standard AFM is incapable of these measurements. However, we do not do SPIM AFM because it has already been established to provide insights similar to those of micromanipulation, and it is technically demanding. The ability to measure and separate these two regimes is vital to the paper as we use it along with modeling to conclude that HP1α is a chromatin crosslinker more than a chromatin-lamin linker in Figure 4. Therefore, further validations of nuclear force response measurements are unnecessary.

For mitotic chromosome measurements, micromanipulation provides novel measurements of single isolated mitotic chromosome mechanics that have yet to be accomplished by other forms of force measurements.

B. The presented measurements demonstrate clear statistically significant differences between WT and HP1α-depleted cells. Our measurements show clear statistical significance change in the chromatin-dominated short extension nuclear spring constant for HP1α degradation in CRISPR HP1α-AID-sfGFP cells (Figure 2, untreated n=13 0.40±0.03 nN/μm vs. auxin n=18 0.22±0.03 nN/μm, t-test p = 0.0002). Oppositely, the strain stiffening lamin-A-dominated nuclear spring constant does not change (Figure 2—figure supplement 1, untreated 0.19±0.05 nN/μm vs. auxin 0.19±0.05 nN/μm, t-test p = 0.998). These measurements (one which changes and one that does not) are taken from the same nucleus force-extension measurements providing strong internal controls. The control parental cell line untreated vs auxin-treated shows an insignificant change (Figure 2 C, untreated n=8 0.35±0.04 nN/μm vs auxin n=11 0.34±0.02 nN/μm, t-test p = 0.72). Also parental U2OS vs. modified HP1α-AID-sfGFP untreated shows an insignificant change (parental untreated n=8 0.35±0.04 nN/μm vs. n=13 0.40±0.03 nN/μm, p = 0.42). Taken together, our data is both unique and rigorously shown to be statistically significant for changes in the chromatin-based nuclear force response regime due to HP1α degradation.

5. The validity of the polymer model that excludes a role for HP1-lamina or inner nuclear membrane interactions in nuclear mechanics is based solely on the short- vs long-extension stiffness data that the authors have previously validated for chromatin vs lamina-dependent mechanical regimes. The role of HP1 and lamina interactions through LBR or PRR14 should be experimentally excluded with mutants or LBR/PRR14 siRNA, in particular as these proteins have been shown to be important for nuclear shape maintenance.

The Reviewer questions the conclusions surrounding the polymer simulations model specific to ruling out HP1α-lamin interactions. The Reviewer is correct that the polymer model uses changes in short vs. long extension to determine whether HP1α degradation or loss of dimerization disrupt chromatin-chromatin or chromatin-lamin interactions. First of all, we would like to point out that the main purpose of the simulations is to investigate the possible mechanical roles of HP1α (as opposed to solely being descriptive of the experimental results). We have updated the manuscript to soften the conclusion drawn from the polymer simulations that vary chromatin-lamina linkages; indeed, the model alone does not rule out HP1α-lamina interactions. However, the model suggests that the changes in mechanical response due to loss of HP1α are accurately modeled as a loss of chromatin-chromatin crosslinkers from a crosslinked chromatin polymer gel. The conclusion of the model is that long-extension strain stiffening should be altered if HP1α has a significant role in the mechanical contribution of chromatin-lamina connections. The experimental data shows no change in the long-extension regime (see Figure 2—figure supplement 1A). Thus, our conclusions that HP1α and its dimer function are primarily functioning mechanically as a chromatin-chromatin crosslinker and not primarily as a chromatin-lamin linker are well supported.

Nonetheless, to address the Reviewer’s comment directly, we considered how HP1α might mediate chromatin-lamina interactions. Previously, PRR14 and LBR have both been shown to interact with HP1α, but neither of which has been shown to affect HP1α localization or behavior. Loss of PRR14 or LBR specifically disrupts peripheral localization of heterochromatin, which is measured by loss of the percentage of H3K9me^2,3^ chromatin at the periphery proximal to the lamina (Poleshko et al., 2013; Dunlevy et al., 2019 biorxiv; Solovei et al., 2013). If loss of HP1α disrupts chromatin-lamina linkages OR either of these proteins, it should phenocopy loss of PRR14 and LBR proteins, and we would expect peripheral heterochromatin enrichment to be disrupted.

We therefore measured localization of HP1α and its substrate heterochromatin near the nuclear periphery. Our new experiments added to the updated manuscript reveal no peripheral enrichment of HP1α and no change in peripheral H3K9me^2,3^ heterochromatin enrichment upon HP1α degradation (Figure 4, E and F). This data suggests that (1) HP1α is not enriched at the periphery like known chromatin-lamina tethers should be [0.86+/- 0.01 peripheral/total average signal], (2) HP1α degradation does not disrupt H3K9me2,3 peripheral enrichment [1.48 +/- 0.04 vs 1.34 +/- 0.03, p> 0.05], which suggests it is not a significant chromatin-lamina tether in this cell line, AND that its loss also does not disrupt LBR or PRR14 needed for maintenance of peripheral heterochromatin, and finally, (3) supports our previous assertion that HP1α’s main mechanical function is best recapitulated in simulations as a chromatin-chromatin crosslinker and not a chromatin-lamina tether.

6. Studies around the role of HP1alpha in mitotic segregation are not described clearly and the connection with chromosome mechanics remain unclear. The experiments are severely underpowered with only very few cells analyzed for mechanical measurements (just 8-14 nuclei) and 3-fold differences in sample size per condition (Figure 5G" "n=15-50 mitotic cells for each condition…") for metaphase misalignment measurements.

We have revised the text to include a clearer explanation of the connection between nuclear mechanics and mitotic segregation. In short, heterochromatin has been shown to have a mechanical role in nuclear mechanics, and, more recently, also a role in mitotic chromosome mechanics (Biggs et al., 2019 MBoC). We launched into similar questions of the differential contributions of HP1α and histone methylation in mitotic chromosome mechanics, with a similar approach as we performed on interphase nuclei. Furthermore, we find that HP1α has a functional role in mitosis, since HP1α degradation disrupts proper metaphase alignment and anaphase segregation. Thus, while a direct link between mechanics and mitotic function remains elusive, we report a close association between perturbations to HP1α-dependent mechanics and function.

Additionally, the Reviewer is concerned about low numbers of measurements. First, we again note that the style of reporting sample size (n) was confusing and we have revised the manuscript to share all the sample sizes of each replicate. To fully address this comment we have new data for both (A) single-chromosome micromanipulation force measurements as well as (B) metaphase and anaphase fidelity measurements. The new measurements further support the original data and conclusions of the paper. Furthermore, we have provided new experiments to quantify HP1αAID-sfGFP presence in isolated chromosome bundles; report HP1α-AID-sfGFP binding mitotic chromosomes in live cells at pericentromeric regions; and measure the effect of abnormal mitosis on interphase nuclear shape of resulting sister nuclei.

A. In the updated manuscript, we now include more single-chromosome micromanipulation force measurements along with measurements of HP1α-AID-sfGFP levels in isolated chromosome bundles and single chromosomes. We have added 6-8 new force measurements of each condition to increase our total micromanipulation chromosome force measurements to 14-20 (Figure 5E).

B. In the updated manuscript we provide new experimental measurements of metaphase alignment and anaphase segregation to bolster the number of events. The revised manuscript provides one more experiment for control (- auxin) and HP1α degradation (+ auxin 16 hours). Furthermore, we added three experiments for rapid degradation of HP1α via + auxin 4 hours, which show an increase of mitotic abnormalities in HP1α-depleted cells, but less than in 16 hours of degradation of HP1α. These data support the main conclusions that HP1α degradation results in mitotic errors, specifically metaphase misalignment and anaphase missegregation.

In the original manuscript, we had provided data from 3 experiments for each condition without or with auxin, where each experiment had at least 15 nuclei. We understand that the original manuscript wording was confusing concerning the true number of measurements. We have now clarified in the figure legends as to the number of experiments and list the number of measurements of each. Overall, this confusion underrepresented the number of measurements.

In the revised version, we have clearly written out all the numbers of cells measured. “Metaphase misalignment 3-4 experiments (black dots) consisting of n = 16, 15, 20, 37 -aux, n = 33, 33, 24 +aux 4 hours, n = 22, 48, 58, 54 +aux 16 hours. Anaphase missegregation 3-4 experiments (black dots) consisting of n = 29, 23, 30, 30 -aux, n = 32, 29, 18 +aux 4 hours, n = 20, 35, 36, 45 +aux 16 hours.” (Figure 5 legend)

Reviewer #2:The manuscript from Strom, Biggs, and Banigan, et al. is an outstanding effort and an important contribution to understanding the complex roles chromatin and protein effectors play in the nucleus. It certainly rises to the level of publication, and I found it to be generally very exciting and compelling work. I have one major criticism of the work below, but would be happy to accept the paper for publication.The major criticism I have of the work is the validation of the HP1 protein after addition of the degron and sfGFP. Especially, given that there are a number of findings in this work are seemingly contrary to expectations, i.e no change in gene expression or H3K9 maintenance. From my reading, the only validation in the manuscript that I see that adding the degron and GFP onto the C-terminus of HP1a (more than tripling the MW right next to the protein-protein interaction hub of HP1) does not affect its biology is a localization measurement. And while I realize that it is not easy to validate a structural protein, it would be nice to have some more corroboration or comparison to experiments that have investigated the formation of heterochromatin by HP1 proteins.

The Reviewer requests more data to support that HP1α-AID-sfGFP functions similarly to unmodified HP1α. We have addressed this through (A) RNA seq; (B) hetero/euchromatin levels; (C) nuclear force measurements (data from the original manuscript) and shape measurements; and (D) mitosis. Oppositely, auxin-induced HP1α degradation does result in changes to nuclear mechanics, shape, and mitotic fidelity.

A. In the revised manuscript, we include RNA seq data showing that there are no significant differences in transcription profiles between parental and AID-sfGFP modified cell line (data supplement). This results in 76 upregulated and 56 downregulated transcripts which represents a 0.8% change across >16,600 genes. GO analysis reveals no change to the nucleus or chromatin proteins suggesting that HP1α-AID-sfGFP does not cause abnormalities associated with chromatin. GO analysis of cellular function returns only extracellular changes for upregulated genes (matrix, region, space, exosome, and vesicle) and no significant enrichment for downregulated genes.

B. Heterochromatin (H3K9me^2,3^) levels were found to be similar between the parental and modified cell lines for immunofluorescence (Figure 2—figure supplement 1D).

C. In the original manuscript, the parental cell line has a nuclear spring constant similar to that of the modified cell line (see Figure 2, p > 0.1) further suggesting that chromatin structure is unaltered by the HP1α-AID-sfGFP modification. In agreement with these results, HP1α-AIDsfGFP untreated nuclei maintain normal nuclear shape that is not significantly different from parental nuclear shape (Parental 0.114 +/- 0.002 μm^-1^; HP1α-AID-sfGFP 0.118 +/- 0.001 μm^-1^, p > 0.05), until HP1α is degraded by auxin (HP1α-AID-sfGFP 0.136 +/- 0.008 μm^-1^, p < 0.0001). Additionally, auxin treatment of the parental line has no effect on nuclear shape in 8 trials (Parental – Auxin 0.114 +/- 0.002 μm^-1^; Parental + Auxin 0.117 +/- 0.001 μm^-1^, p > 0.05). Abnormal nuclear shape upon HP1α degradation is rescued by expression of an exogenous wildtype HP1α (see Figure 3 in main text). Taken together, modified HP1α-AID-sfGFP maintains chromatin-based nuclear mechanics and shape, similar to wild-type HP1α.

**Author response image 3. sa2fig3:** 

D. In new experiments, we verify that parental mitosis fidelity is similar to the modified cell line (Figure 5G). Upon degradation of HP1α-AID-sfGFP mitotic failures increase.Altogether, these different experiments show that the nuclear functions that are disrupted by auxin-induced degradation of HP1α-AID-sfGFP are not perturbed by the presence of HP1α-AIDsfGFP in place of WT HP1α. This supports the functionality of the HP1α-AID-sfGFP protein construct.

For example, to reconcile these data with previous findings, one might predict that gene expression or H3K9 levels should eventually recapitulate previous knockdown/out studies if the auxin treatment were to persist, provided cell type and cell cycle considerations. While RNA seq experiments are fairly involved, I would think it possible to test the methylation levels in the auxin-HP1 line after a few days of depletion to see if levels do come down to previously reported levels, at least at specific satellite sequences. Alternatively, it might be easier to assay the AID-GFP-HP1 at exogenous sites as has been previously reported. Though, it is noted however that the dimerization mutant does perform as expected.In addition to validation, it would also be helpful to more directly compare the timing of your experiments with those done elsewhere to give the reader a better idea of when changes (epigenetic, transcriptional) are likely to arise, etc. Though, discussion is somewhat confounded by the diversity of HP1 activity in different organisms. There is the distinction of >24 hours in the discussion, but there are several HP1 experiments that show faster changes than this, specifically compaction following recruitment and transcriptional repression at exogenous sites.

The Reviewer requests that as an additional validation, we do long-term loss studies in addition to short-term loss (4 hours of auxin treatment), which we focused on in the original manuscript. This is indeed an interesting question, and we have provided new data on this point. In the revised manuscript, we show that histone methylation levels are not changed at 16 hours of auxin treatment and 96 hours (Figure 2—figure supplement 1, D and E). Additionally, in the original manuscript, we provided the raw data showing lack of transcriptional change after 16 hours of auxin treatment. In the revised manuscript, we note the lack of change in transcription and have attached the RNAseq data as a data supplement. We were unable to do longer term loss experiments (>24 hours) for transcription and satellite analysis due to the technical challenges. Finally, we have revised the manuscript to further discuss how HP1α functions differently in different organisms (Li et al. 2003; Verschure et al. 2005; Fischer et al. 2009; Sadaie et al. 2008; Lee et al. 2019; Levine et al. 2015; James and Elgin 1986; Schwaiger et al. 2010; Singh et al. 1991; Wreggett et al. 1994)

Reviewer #3:The authors use state-of-the-art genome engineering and degron tagging approaches combined with transcriptional profiling and sophisticated biophysical measurements. Most of the results are clear and convincing, but the section on mitotic chromosomes has substantial weaknesses and should either be removed or extended to substantiate the key conclusions. The main part on HP1alpha's role in interphase nuclei, however, is interesting and well supported by the data.

We appreciate that the Reviewer feels that the work on interphase nuclear mechanics and shape is “clear and convincing” and “interesting and well supported”. We have worked to revise the mitotic section (Figure 5) to rise to this level. We believe that our substantial amount of new experiments address these three major concerns raised by the Reviewer. We believe that the improved mitosis data will be equally interesting to the broad readership of *eLife*.

1. Localization of HP1alpha on mitotic chromosomes. The authors claim that HP1alpha remains bound to mitotic chromosomes, but this is not visible in the data shown in the paper. Prior studies have in fact clearly shown that HP1alpha completely dissociates from chromosome arms, while only a small fraction remains bound to pericentric chromatin (Hirota et al., 2005; Fischle et al., 2005; Serrano et al., 2009; Akram et al., 2018). The authors reference these papers to provide a rationale for studying HP1alpha's potential role in mechanical rigidity of chromosome arms, but they neglect the fact that the previous studies showed that HP1alpha only localizes to a very small region surrounding centromeres. This very limited enrichment of HP1alpha at pericentromeric regions is inconsistent with a direct major mechanical role of HP1alpha in stiffening chromosome arms as the authors suggest.Figure 5A shows HP1alpha-GFP fluorescence throughout the entire cell with a potentially very slight enrichment in the central region. Without counterstaining DNA or chromatin, it is not possible to assess whether any HP1alpha remains bound to chromosomes, as chromosome positions are not known. The authors should provide high-resolution figures of HP1alpha-GFP counterstained with a DNA dye, ideally in live cells. They should further image chromosome spreads as in the prior work by Serrano et al., 2009 to clarify where on chromosomes HP1alpha binds. Figure 5C shows a fuzzy fluorescence signal for HP1alpha, but the image quality is insufficient to assess whether this is on chromosome arms or on pericentromeric regions. Moreover, without a control cell that does not express GFP-tagged HP1alpha, it is not possible to assess to which extent the fuzzy fluorescence signal shown in Figure 5C represents background autofluorescence. Alternatively, as the mitotic chromosome stiffness measurements are also not very convincing (see next point), the authors might consider removing the entire section on mitotic chromosomes.

The Reviewer requests data supporting the presence or absence of HP1α at both the centromere and on chromosome arms and how HP1α location relates to mechanics. New data show that HP1α is located throughout the chromosome and concentrated at the pericentromeric region. Specifically, to address this comment we have included: (A) Imaging of live ex vivo isolated mitotic chromosome bundles by widefield imaging that shows HP1α fluorescence as concentrated pericentromeric foci as well as a weaker signal on chromosome arms, see Figure 5 B, C, —figure supplement A and B; (B) Live-cell imaging of metaphase chromosomes by confocal, which recapitulates HP1α’s enriched localization to pericentromeric foci, whereas Hoechst-stained DNA is equally distributed, see Figure 5—figure supplement C; (C) Revisions to the text to clarify that HP1α is largely located at the centromere in our experiments, but can nonetheless contribute to whole-chromosome mechanics, and moreover, that micromanipulation force measurements are not solely a measure of chromosome arms.

A. We have revised the manuscript and Figure 5 to include several examples of ex vivo isolated mitotic chromosome bundles that clearly show HP1α-AID-sfGFP on the mitotic chromosomes. These images show dense foci, likely at the centromere, as well as a lesser, but still detectable, presence on the chromosome arms (Figure 5 B and C). We have also provided controls for autofluorescence and HP1α auxin-induced degradation (see Figure 5—figure supplement 1). We note in the revised manuscript that the cellular background is bright, possibly due to some amount of unbound HP1α, which could disrupt the ability to see low levels of HP1α on mitotic chromosome arms. By removing the chromosome bundle from the cell this decreases the background, making HP1α-AIDsfGFP visible on chromosomes throughout their arms. Thus, our ability to remove cellular background through chromosome bundle isolation is sufficient to reveal HP1α on the chromosome arms. We would also like to note a few points about the mentioned references:

– Hirota et al., 2005 imaging in cells – subject to background fluorescence

– Fischle et al., 2005 imaging in cells – subject to background fluorescence

– Serrano et al., 2009 chromosome spreads of HeLa cells show more than pericentromeric foci; they also show multiple foci across the chromosome (Figure 1). Thus, this work agrees with our findings that there is HP1α on the chromosome arms (although there is clearly less of it). We thank that Reviewer for pushing us to more carefully consider the literature and to do better fluorescence measurements. Specifically, the Reviewer suggested chromosome spreads. While we chose to use a different method, Serrano et al. provide initial data that chromosome spreads show HP1α at both the pericentromeric region and on the arms. This data is in agreement with our findings using chromosome bundle isolation.

– Akram et al., 2018 imaging in cells – subject to background fluorescence, though not all HP1α fluorescence appears to colocalize with pericentromeric labeling via ACA (see Figure 6).

– Overall these cited publications show that Aurora B facilitates HP1α removal from chromosome arms, suggesting that HP1α does have the capacity to bind to both the pericentromeric and chromosome arms. However, a measurable level of remaining HP1α may be obscured inside cells given strong cellular background fluorescence. In Figure 5 and Figure 5—figure supplement 1, A and B, we address the Reviewer’s worries about levels of background fluorescence and now show that isolated chromosome bundles have less background fluorescence, allowing us to better visualize and measure HP1α presence and localization.

B. Live-cell imaging using confocal microscopy shows HP1α-AID-sfGFP as foci corresponding to centromeres relative to DNA staining of whole chromosomes (Figure 5—figure supplement 1C). This data is in agreement with cited papers in the original manuscript as well as the papers referenced by the Reviewer.

C. We have clarified in the manuscript that micromanipulation force measurements of an isolated mitotic chromosome include the whole chromosome, arms and the centromere, and does not solely measure chromosome arm mechanics. The revised manuscript now states: “The resistive force measured includes both HP1α on the arms and at the pericentromere since tension is distributed across the whole chromosome from the pipettes holding opposite ends of the chromosome (Figure 5D example images).” Additionally, recently published data of in vitro work with HP1α demonstrates that in an isolated system, accumulation of HP1α in small puncta on DNA is sufficient to contribute to the mechanics of the entire DNA fiber (Keneen et al., *eLife* 2021). Thus, even beyond our finding of HP1α on arms, this suggests that accumulation of HP1α at pericentromeric sequences could contribute to mechanics of the entire chromosome.

We have novel evidence for HP1α presence on chromosome arms via chromosome bundle isolation and imaging. Furthermore, our data agrees with many other publications that HP1α remains at the pericentromere during mitosis. Micromanipulation of an isolated chromosome stretches the chromosome from end-to-end while measuring force vs. extension throughout the chromosome from one arm through the centromere to the other arm. Thus, HP1α located in the arm as well as the pericentromere would allow it to resist stretching in our micromanipulation force measurement experiments. Previous studies imaging condensin along the chromosome during micromanipulation stretching provide preliminary data that the centromere stretches less than chromosome arms (Sun et al., Chromosome Res 2018). Thus, it may be possible that the changes in mechanics are dependent on HP1α’s mechanical role at the centromere. However, our newly added data of HP1α-AID-sfGFP chromosome bundle imaging (see point A above) supports that HP1α is present at chromosome arms as well. Further studies will be needed to untangle the mechanical and structural role of the chromosome arms vs. the pericentromere, but our novel data provides new and exciting insights.

2. HP1alpha's contribution to mechanical rigidity of mitotic chromosomes. The authors claim that HP1alpha depletion "strongly impairs" mitotic chromosome mechanics. Their microneedle-based force extension experiments shown in Figure 5E, however, show a large variability between individual measurements but only a very slight difference between HP1alpha-AID-sfGFP cells treated with auxin compared to untreated cells. There is a star that might indicate statistical significance (this is not explained in the legend), but even if that were true then the effect size would still be so small that biological relevance is questionable. Given the concerns about HP1alpha localization, the overall evidence supporting a "major role" of HP1alpha in the stiffness of mitotic chromosomes is insufficient to support this conclusion. The authors might either perform additional experiments to substantiate the conclusions or remove the part on mitotic chromosomes.

The Reviewer requests clarification and more data to support the major conclusion that HP1α is a mechanical component of mitotic chromosomes. To directly address this comment, we have revised the text to use more measured language, specifically removing “major” and instead saying “a mechanical component of the mitotic chromosome”. Furthermore, as requested by the Reviewer, we have provided 6-8 additional force measurements to increase the total number measurements to 14-20 for each scenario. These new measurements provide more data to support the conclusion that HP1α is a mechanical component of mitotic chromosomes as auxin-degradation decreases the doubling force from 262 +/- 50 to 148 +/- 12 pN which is statistically significant (n = 20 each, p = 0.03) and represents an effect of size of ~40%. We have also revised the figure legends to clearly indicate “P values reported as * < 0.05, **< 0.01, ***< 0.001.”

The Reviewer pointed out that the force measurements show large variation between individual chromosome measurements. Variability likely stems from the inability to select a specific chromosome (random selection of 1 of 23 chromosomes), isolation of the chromosome, holding of the chromosome, and inherent variability of the mechanical measurement technique. Recent advancements in the technique have aided isolation efforts of a single chromosome by using a third micropipette to hold the bundle of chromosomes while the other two hold the ends of the single chromosome, allowing the ends to be pulled away from the bundle together. This technique aids maintaining the single chromosomes’s size and shape while it is pulled away from the bundle. Furthermore, we use doubling force to account for variability in which chromosome is grabbed from the bundle and where precisely the chromosome is grabbed at either end. This mitages issues of how length (and thickness) of the chromosome can affect strength, since a longer spring is weaker than a shorter spring given a set extension, and normalizes for these differences.

Overall, we believe our data is consistent, which is further aided by newly added data of 6-8 more measurements that ultimately supports that loss of HP1α results in a weaker mitotic chromosome. Furthermore, the new data help support that increased histone methylation levels via methylstat are dominant in chromosome mechanical stiffness as compared to HP1α since methylstat + auxin treatment does not significantly differ from methylstat treatment alone (auxin and methylstat 452 ± 116 pN vs. methylstat 745 ± 164 pN, p = 0.17).

3. HP1alpha's role in regulating nuclear shape. The authors suggest that the softening of interphase nuclei resulting from degradation of HP1alpha directly causes aberrant nuclear morphologies. However, they also show that depletion of HP1alpha leads to chromosome missegregation, which is known to lead to aberrant nuclear morphologies owing to perturbed nuclear assembly during mitotic exit. To assess to which extent the nuclear morphology defects in HP1alpha depleted cells are caused directly via nuclear softening or indirectly via chromosome missegregation, the authors should separately analyze nuclear morphology shape changes as shown in Supplementary Figure 2 for those cells that have passed through mitosis after HP1alpha degradation versus those cells that remained in interphase during the entire imaging duration.

The Reviewer requests experiments to address how abnormal mitosis affects nuclear shape in interphase. We have original manuscript data as well as new experiments to clearly address this important question.

First, we would first like to point out that in the original manuscript, increased nuclear curvature in interphase was shown to proceed coincident with HP1α degradation and independent of mitosis (see Figure 2—figure supplement 2). This original data addresses the issue of whether cells with abnormal nuclei have passed through mitosis “versus those cells that remained in interphase during the entire imaging duration”. Cells and their nuclei were tracked over 12 hours in interphase for nuclear curvature in auxin -/+ treated. This tracking shows loss of nuclear shape, measured as increased nuclear curvature over the entire tracked population (panel E), coincident with loss of HP1α-AID-sfGFP fluorescence signal at around 4-6 hours. Histograms also reveal the major shift in individual nucleus curvatures occurs within the 4-hour time window (panels D). This is the same time interval in which we measure nuclear mechanical softening (Figure 2D).

To directly address this Reviewer's comment we provide new data in the revised manuscript tracking cells through mitosis and into interphase after 24-hour pre-treatment in auxin. Our new experiments reveal that upon auxin-induced HP1α degradation both normal and abnormal mitosis result in daughter cells that have increased nuclear curvature (Figure 5G). In cells with degraded HP1α, abnormal mitoses are much more common, as we reported earlier, and these abnormal mitoses also lead to mostly abnormally shaped daughter nuclei. However, even cells that go through normal mitoses have higher average curvature than those in the “- Auxin” case, suggesting that abnormal mitosis is not required for abnormal daughter cell shape. Taken together, this data supports that rapid degradation of HP1α results in abnormal nuclear morphology (high nuclear curvature) both during interphase independent of mitosis AND over longer time intervals (16 hours) sister nuclei post mitosis independent of normal vs abnormal mitosis.